



# Evolution of an extreme Pyrocumulonimbus-driven wildfire event in Tasmania, Australia

Mercy N. Ndalila[1], Grant J. Williamson[1], Paul Fox-Hughes[2], Jason Sharples[3] and David M. J. S. Bowman[1]

[1] School of Natural Sciences, University of Tasmania, Hobart, Tas. 7001, Australia
[2] Bureau of Meteorology, Hobart, Tas. 7001, Australia
[3] School of Science, University of New South Wales, Canberra, ACT 2601, Australia

*Correspondence to*: mercy.ndalila@utas.edu.au

**Abstract.** Extreme fires have substantial adverse effects on society and natural ecosystems. Such events can be associated with intense coupling of fire behaviour with the atmosphere, resulting in extreme fire characteristics such as pyrocumulonimbus cloud (pyroCb) development. Concern that anthropogenic climate change is increasing the occurrence of pyroCbs globally is driving more focused research into these meteorological phenomena. Using 6-minute scans from a nearby weather radar, we describe the development of a pyroCb during the afternoon of 4 January 2013 above the Forcett-Dunalley fire in south-eastern Tasmania. We relate storm development to: (1) near-surface weather using the McArthur Forest Fire Danger Index (FFDI), and the C-Haines Index, a measure of the vertical atmospheric stability and dryness both derived from gridded weather reanalysis for Tasmania (BARRA-TA), and (2) a chronosequence of fire severity derived from remote sensing. We show that the pyroCb rapidly developed over a 24-minute period in the afternoon of 4 January, with the cloud top reaching a height of 15 km. The pyroCb was associated with a highly unstable atmosphere (C-Haines 10-11) and Severe-marginally Extreme (FFDI 60-75) near-surface fire weather, and formed over an area of forest that was severely burned (total crown defoliation). We use spatial patterns of elevated fire weather in Tasmania, and fire weather during major runs of large wildfires in Tasmania for the period 2007-2016 to geographically and historically contextualise this pyroCb event. Although the Forcett-Dunalley fire is the only known record of a pyroCb in Tasmania, our results show that eastern and south-eastern Tasmania are prone to the conjunction of high FFDI and C-Haines values that have been associated with pyroCb development. Our findings have implications for fire weather forecasting and wildfire management, and highlight the vulnerability of southeast Tasmania to extreme fire events.

**Keywords:** fire weather; fire severity; extreme fire; C-Haines; McArthur Forest Fire Danger Index; smoke plume injection; pyrocumulonimbus; Tasmania; wildland fire.

## 1 Introduction

Anthropogenic climate change is increasing the occurrence of dangerous fire weather conditions globally (Jolly et al., 2015; Abatzoglou et al., 2019), leading to energetically intense wildland fires. For instance, climate projections suggest a pronounced



increased risk of extreme fire events in Australia, with 15-70% increase in number of days conducive for extreme wildfire by 2050 in most locations (Hennessy et al., 2005), although the models show inconsistencies in the trajectory and variability of future fire weather, especially in eastern and south-eastern Australia (Clarke and Evans, 2019; Clarke et al., 2011). While fire

weather is most often understood as a surface phenomenon; for example, through surface temperature, wind speed and fuel dryness, atmospheric instability can also drive extreme fire development. Extreme wildfires have been defined as fires that exhibit deep flaming and are coupled to the atmosphere, resulting in the development of violent pyroconvection (Sharples et al., 2016). In some cases, violent pyroconvection can manifest as pyrocumulonimbus clouds (pyroCb), the tops of which can reach the upper troposphere and lower stratosphere, and inject aerosols into those altitudes where they can be transported for

thousands of kilometres, and hence affect global climate (Trentmann et al., 2006; Fromm et al., 2010; Peterson et al., 2018).

PyroCbs lead to highly erratic fire behaviour because of strong updrafts and downdrafts, the possibility of associated whirlwinds and tornadoes, and rapid fire growth due to heightened ember generation, long-range spotting and cloud to ground lightning strikes (Lareau and Clements, 2016; Dowdy and Pepler, 2018; Fromm et al., 2010; Tory and Thurston, 2015;

Cunningham and Reeder, 2009). Specifically, downdrafts can cause erratic fire spread, driven by sudden wind gusts impacting the surface in multiple directions, endangering firefighters near the pyroCb (Johnson et al., 2014; Potter and Hernandez, 2017) and frustrating accurate prediction of fire behaviour. Local surface processes can amplify fire behaviour, for instance, eddies in steep lee-facing slopes can cause lateral fire spread and mass spotting in downwind areas, a process known as fire channelling or vorticity-driven lateral spread (VLS) (Sharples et al., 2012).


PyroCbs are comparatively poorly understood meteorological phenomena as until recently they were historically rarely observed or studied world-wide (Fromm et al., 2010). In Australia, pyroCbs have been confirmed for at least 65 fire events (R. McRae, Pers. Comm.), with more than one event occurring over a single fire in some instances. The most significant events have produced plumes that have reached the upper troposphere/lower stratosphere (UTLS) region. Insights into pyroCb

development have relied on weather radar (Melnikov et al., 2008; McCarthy et al., 2019; Peace et al., 2017; McRae, 2010; Johnson et al., 2014; Fromm et al., 2012) that provides high temporal resolution imagery of the pyroconvection, although importantly they do not accurately detect the exact extent of entrained gaseous and fine particulate emissions because they are 'tuned' to identify larger particles such as rain and ice crystals, and can therefore fortuitously detect ash and embers.

An important correlate of pyroCb formation is atmospheric instability and moisture (Luderer et al., 2006; Lareau and Clements, 2016; Rosenfeld et al., 2007; Fromm et al., 2010; Di Virgilio et al., 2019). A fire weather index commonly used in Australia to monitor meteorological conditions in the lower atmosphere is the continuous Haines Index (C-Haines; Mills and McCaw, 2010). The index is a modification from the Haines index (Haines, 1988) which is routinely used in the US, but adapted to suit the frequent hot and dry summer conditions in Australia. The C-Haines index provides a measure of the potential for erratic

fire behaviour, based on temperature lapse and moisture content between two lower atmospheric levels, and ranges from 0-13.


High values of C-Haines imply drier and more unstable atmospheric conditions, which favour lifting of the heated air higher into the atmosphere. In particular, a large temperature lapse in the atmosphere favours the maintenance of strong convection and increases the likelihood of pyroCb development.

Here, we describe the evolution of a pyroCb event in south-eastern Tasmania that developed on 4 January 2013 during the Forcett-Dunalley fire (Ndalila et al., 2018). We use the Mt. Koonya weather radar to document the temporal evolution of the pyroCb and relate storm development to near-surface fire behaviour using the McArthur Forest Fire Danger Index (FFDI), and vertical atmospheric stability and dryness using the C-Haines Index, both derived from gridded weather reanalysis for Tasmania (BARRA-TA). To understand how fire behaviour may have influenced the storm, we also associated the

development of the pyroCb with a map of the temporal progression of fire severity derived from remote sensing and field observations (Ndalila et al., 2018) and terrain analysis based on a digital terrain model. Finally, we contextualise the pyroCb event by determining: (i) the FFDI and C-Haines associated with large wildfires in Tasmania within the period covered by the available BARRA meteorological reanalysis (2007-2016); and (ii) the geographic patterns of days with concurrent elevated values C-Haines and FFDI in Tasmania.

**2 Methods**

**2.1 Study area**

The Forcett-Dunalley fire occurred on the Forestier and Tasman Peninsulas in the southeast of Tasmania (Fig. 1a), a temperate island state to the south of Australia (Fig. 1a). This region has a cool moist climate, and elevation reaching 600m above sea level (Fig. 1d). The fire was reported at 14:00 local time on 3 January 2013, ignited from a smouldering stump from an

unextinguished campfire.  On 4 January, southeast Tasmania recorded dangerous fire weather conditions, resulting in a large uncontrollable fire that led to the development of a pyrocumulonimbus (Fig. 1b) and caused the near-complete destruction of Dunalley township (Fig. 1d). By the time of containment on 18 January, the fire had burnt 20,200 ha, mostly comprising native vegetation and rural lands (Fig. 1c). A detailed description of the fire and associated broader meteorological and environmental conditions have been provided in previous reports/studies (Bureau of Meteorology, 2013; Ndalila et al., 2018; Fawcett et al.,

2014; Marsden-Smedley, 2014).









**Figure 1:** Location of the Forcett-Dunalley fireground in SE Tasmania: (a) Annual rainfall (in mm) and elevation (in metres)
across Tasmania, and the location of major fires in the 2013 fire season including Forcett-Dunalley (1); (b) Photographs of a
pyrocumulonimbus cloud above Dunalley township, taken in the afternoon (around 15:55) of 4 January 2013. (c) Dominant
vegetation in the Forestier and Tasman Peninsulas based on *TASVEG 3.0*, an integrated vegetation map of Tasmania.
*Eucalyptus* (Euc.) is the major vegetation type in the region. (d) Elevation, and mean annual rainfall (dotted lines) across the
two Peninsulas, derived from *Worldclim* dataset (Hijmans et al., 2005). The locations of Dunalley township and Mt. Koonya
weather radar are indicated on the map.

## 2.2 Meteorology of the fire

### 2.2.1 PyroCb development

We used weather radar to track the evolution of the pyroCb during the period of erratic fire behaviour. Since most weather
radars do not detect all components of a smoke plume, especially particles smaller than 100 µm (Jones and Christopher, 2010),
we will refer to the signatures present in the radar data as 'the plume', which will also encompass the pyroCb
cloud. Pyroconvective plumes typically contain some precipitation (and in extreme cases, glaciation from pyroCb) at high
altitudes, as well as larger smoke particles (some of which can be water-coated). In this study, we assume the highest radar
returns (plume injection) during the period of violent convection represent the top of the pyroCb. This is justified as McRae et
al. (2015) also show that stronger radar returns at the highest altitudes during deep pyroconvection events are mostly from
hydrological features (ice crystal and rain) rather than non-hydrological components (ash and debris).

We examined radar data for the period 3-18 January 2013 from the Mt Koonya C-band (5 cm wavelength) weather radar (Fig.
1d) operated by the Australian Bureau of Meteorology (Soderholm et al., 2019). The radar's proximity and uninterrupted view
of the Forcett-Dunalley fireground made it ideal for tracking the pyroCb. The radar is located at a height of 515 m, 46 km east-
southeast of Hobart on the Tasman Peninsula, and its scan strategy contains 14 elevation angles, scanning through 360° within
each angle. The radar is 24 km from Dunalley, where the lowest (0.5°) elevation scan height is about 750m, while the highest
scan (32°) is around 25 km elevation at a distance 40 km from the radar. The radar has a one-degree beam width, 250 m radial
resolution, an effective range of 150 km and primarily provides 6-minute reflectivity and velocity scans of the plume/pyroCb.
In this study, a minimum reflectivity of 11 dBZ was used to detect plume boundary. The time zone used in this paper is
Australian Eastern Summer Time, referred to as local time (LT) in this paper.

Three-dimensional radar scans from 12:30 to 23:00 LT on 4 January were used in describing the pyroCb as they represent the
period of peak fire behaviour. In the later periods, smoke had considerably reduced and was not visible on the radar. Radar
files were first processed by converting the raw polar coordinates to Cartesian coordinates. We used the Integrated Data Viewer
(IDV) (Unidata, 2018 ) and ArcGIS 10.3 (ESRI, 2015) to detect and analyse radar returns in 2D and 3D displays, where the
2D plan view of the radar indicates the horizontal extent of the dense plume which includes embers, ash and the pyroCb.
Within IDV, a vertical cross section through the 3D plumes was used to estimate the maximum injection height (cloud top) at


each 6-minute timestamp of the radar data. ArcGIS software was used to measure the horizontal length, and size (area and perimeter) of the 2D view of the plume, based on the lowest elevation angle of the radar. We compared the temporal variation

of plume/cloud development with the time series of mapped fire severity and progression of fire weather (FFDI and C-Haines) to determine any congruence between pyroCb dynamics and fire weather, area burnt and fire severity patterns during the period of erratic fire behaviour.

### 2.2.2 C-Haines analysis

We obtained gridded weather reanalysis data from the Bureau of Meteorology Atmospheric high-resolution Regional Reanalysis for Australia (BARRA), downscaled for the Tasmanian sub-domain (BARRA-TA) to 1.5 km spatial resolution (Su et al., 2019). BARRA combines numerical weather forecasts with observational data to produce realistic depictions of surface meteorology and atmospheric conditions. We extracted hourly air temperature and dewpoint temperature at different air pressure levels (1000 hPa at the surface to 10 hPa in the mid-stratosphere), as well as pre-calculated hourly McArthur FFDI

for the period of the fire and the period of BARRA data available at the time of the study (January 2007-October 2016). Extraction, conversion and general BARRA analysis was performed in R version 3.4.0 software (R Core Team, 2017).

At each grid cell in Tasmania, the C-Haines index was calculated from the hourly estimates of air temperature and dewpoint temperature at relevant atmospheric levels based on Eqs. 1-3. We preferred the BARRA product to radiosonde data to calculate

C-Haines because of: (1) a possible geographic drift of the weather balloon as it rises through the atmosphere, resulting in inconsistencies in locations where data were recorded; (2) availability of balloon data only twice a day at a single location, which is unrepresentative of many regions; and (3) the BARRA product combines other data sources such as satellite observations to model air temperature and moisture. Nevertheless, we validated our C-Haines values by using the radiosonde data for Hobart Airport, establishing a correlation of 0.74 between the two datasets. Our calculated hourly C-Haines values, as

well as the extracted hourly FFDI for Tasmania were then aggregated to maximum daily values. Temporal maps of daily C-Haines distribution were then produced for the first three days of the fire, between 3-5 January.

$$CA = (T850 – T700)/2 – 2 \tag{1}$$

$$CB = (T850 − DT850)/3 − 1 \tag{2}$$
if $(T_{850} - DT_{850}) > 30$, then $(T_{850} - DT_{850}) = 30$
if $CB > 5$, then $CB = 5 + (CB-5)/2$

$$CH = CA + CB \tag{3}$$






where CA is a temperature lapse term; $T_{850}$ is temperature at 850 hPa atmospheric height; $T_{700}$ is temperature at 700 hPa height; $DT_{850}$ is the dewpoint temperature at 850 hPa height; CB is a dewpoint depression term; and CH is the continuous Haines Index (or C-Haines).

### 2.2.2 C-Haines analysis

We also determined whether the period of rapid cloud/plume development coincided with local surface dynamics, which likely enhanced fire behaviour. Specifically, the effect of Vorticity-driven Lateral Spread (VLS) on fire behaviour was tested. VLS-prone areas were defined according to Sharples et al. (2012) criteria as follows: lee-facing slopes steeper than 15°, with slopes facing to approximately 40° of the direction the wind is blowing towards. A wind direction layer (mostly north-westerly) at 16:00 on 4 January (the period around peak plume height) was extracted from the BARRA dataset and resampled to correspond to the spatial resolution of the Digital Elevation Model (33 m), provided by the Tasmanian Department of Primary Industries, Parks, Water and Environment. Both layers were combined using the aforementioned criteria, resulting in a binary map where areas fulfilling the criteria of VLS were assigned a value of one and all other areas zero.

### 2.4 Spatio-temporal context of fire weather in Tasmania

#### 2.4.1 Weather conditions during large Tasmanian fires

We compared FFDI and C-Haines values during the Forcett-Dunalley fire and values associated with other large Tasmanian fires between years 2007 and 2016 (the period of available BARRA data). A total of 77 fires, of varying ignition sources, were identified in the Tasmania Fire Service fire history database as being >500 ha in size, and having a known ignition date. Of these, 18 did not have recorded end dates, and these were operationally specified as being four weeks later, an arbitrary cut-off to capture the most likely major growth (or 'run') of the fire that typically happens at or near the start of fires. We subsequently produced a scatterplot of highest daily FFDI for the duration of each of the fires and the associated C-Haines value. These data were overlaid on a density map of C-Haines and FFDI values for all cells (and days) in Tasmania during the entire BARRA period to provide background weather conditions for Tasmania.

#### 2.4.2 Elevated fire weather in Tasmania

To provide a geographic context to the Forcett-Dunalley fire, the spatial patterns of days conducive for extreme fire behaviour in Tasmania were mapped by determining counts of days that exceeded combined C-Haines and FFDI thresholds of 9 and 25 respectively, then aggregating them to fire season (October-March) and non-fire season (April-September). These thresholds were chosen to define an elevated fire weather day (95[th] percentile) based on weather conditions at Hobart Airport, which





represents the broader airmass in south-eastern Tasmania. For example, the FFDI threshold is close to the 95[th] percentile (FFDI 31) observed during the 1998-2005 fire seasons (Marsden-Smedley, 2014). We chose to use FFDI to describe near-surface fire weather across the state, but acknowledge that for some fuel types (*e.g.* moorlands) occurring in Tasmania (Marsden-Smedley and Catchpole, 1995), other indices such as the Grassland/Moorland Fire Danger Index may be more suitable. It is worth

noting that maximum daily FFDI calculated at the Hobart Airport station was always higher than daily gridded FFDI extracted from the BARRA product.

## 3 Results

### 3.1 Fire weather during the fire

### 3.1.1 Surface fire weather

Dangerous fire weather conditions in southeast Tasmania were observed from 3-4 January 2013. Hobart Airport, at 14:00 LT on 3 January when the fire was reported, recorded 33 ℃ (the maximum temperature for the 3[rd]), relative humidity (RH) of 15 %, and strong north westerly winds reaching 37 km h$^{-1}$ and gusting to 55 km h$^{-1}$ (Fig. 2). The smoke plume was detectable by weather radar between 15:18 and 19:00 LT. The weather conditions deteriorated on 4 January, with extreme maximum temperatures (reaching 40 ℃), strong winds (35-46 km h$^{-1}$, gusts of 60-70 km h$^{-1}$), and low RH (11 %) in the afternoon. At

12:00, the Forcett-Dunalley plume was again detectable by weather radar (Figs. 2 & 3a), when values of the weather variables peaked and were maintained during the period of violent pyroconvection between 15:30-16:30 (Fig. 2).





**Figure 2:** Time series of 30-minute weather data obtained from 3-4 January 2013 at Hobart Airport. Units: rainfall (mm), Relative Humidity (%), temperature (°C), Wind gust and speed (km h⁻¹). The graph includes the time of the start of the fire and the period of the pyroCb event. The period of the radar detection of the plume for the two days are also shown on the graph.

### 3.1.2 PyroCb development in the atmosphere

On 4 January, the plume height from radar scans gradually increased from around 1 km at 13:00 to 8 km at around 15:00 and rapidly rose to the maximum injection height of 15 km (lower stratosphere) between 15:30 and 15:48 (Fig. 3a), representing the peak of pyroconvection. During the period of violent pyroconvection, thunderstorms developed and moved in a south-easterly direction towards the Tasman Sea, causing two lightning strikes, which were detected around 16:10 (Bureau of Meteorology, 2013). Radar returns during the peak period were likely due to glaciation within the pyroCb and precipitation at high altitudes, which then evaporated before reaching the surface due to intense heat, while the cooled air mass from evaporation descended towards the surface as convective outflows, generating gusty and erratic surface winds. The cloud top




then decreased to around 7 km at 16:42, after which the pyroCb likely dissipated and the plume subsequently stabilised at heights of 3 km. Atmospheric instability and the pyroCb dynamics are confirmed by atmospheric soundings from BARRA for 15:00-17:00 on 4 January (Fig. S1), and a time series of 850-500 hPa temperature lapse rate on 4 January (Fig. S2). The soundings show air temperature at the tropopause (~12 km, below the pyroCb height) to be -50 °C, supporting the observation

of electrification of the pyroCb. Electrification/lightning typically occurs at temperatures around or below -20 °C (Williams, 1989).

**Figure 3:** Plume dimensions, FFDI and fire severity traces during the evolution of a pyroCb on the afternoon of 4 January (a)
Six-minute variation of plume dimensions during peak fire behaviour, where plume height has been scaled for visualisation by multiplying by a factor of 10. The asterisk (*) represents the period (16:42-17:06) with missing weather radar data. (b) FFDI trace obtained from Hobart Airport weather station during the corresponding period of smoke plume growth. (c) Temporal pattern of fire severity, adapted from Ndalila et al. (2018). Black vertical lines in the three graphs represent period of violent pyro-convection.






For the horizontal plume dimensions (length, area and perimeter), there was a lagged response to the effect of intense fire
activity that occurred at around 15:30. While a maximum in cloud height was observed at 15:48, other plume metrics peaked
at around 17:13 (Fig. 3a).

The period of drastic increase in the height of the pyroCb cloud was associated with elevated gridded FFDI of 60-75 (Severe
(50-74) to marginally Extreme (75) fire danger classes), elevated C-Haines of 10-11.1 (Fig. 3b), a large area burnt
(approx.10,000 ha) and the highest proportion of burning of the two highest fire severity (total crown defoliation) categories
(Fig. 3c) in dry *Eucalyptus* forests. The effect of Vorticity-driven Lateral Spread on fire behaviour was observed through plume
evolution during the peak period, as a southwest (lateral) expansion of the upwind edge of the smoke plume by 2 km in just
40 minutes, perpendicular to the prevailing northwest winds (Fig. S4). The majority of the plume area (>70%) extended over
the Tasman Sea in a south-easterly direction from the location of the fire, under the influence of above-surface winds (Fig.
S3).

### 3.1.3 Spatio-temporal variation of C-Haines

During the development of the pyroCb, maximum daily gridded C-Haines and FFDI on 4 January at Hobart Airport were
consistently high (11.1 and 92, respectively). In the following days both these indices markedly declined (Fig. S5); indeed, for
the entire month of January 2013, there was a good correlation of 0.5 between these fire weather indices, although FFDI lagged
C-Haines by around a day. At a state-wide scale, from 3 to 4 January, the whole of Tasmania displayed dangerous fire weather,
particularly south-eastern Tasmania, which recorded high levels of C-Haines (Fig. 4). In the Tasman Peninsula, especially, C-
Haines was mostly in the range of 10-12 for both days (peaking at 12-14) but moderated to 4-6 on 5 January after a southwest
wind change at around 00:00. A complete description of meteorological conditions, including the aerological diagrams, can
be found in the supplementary section.


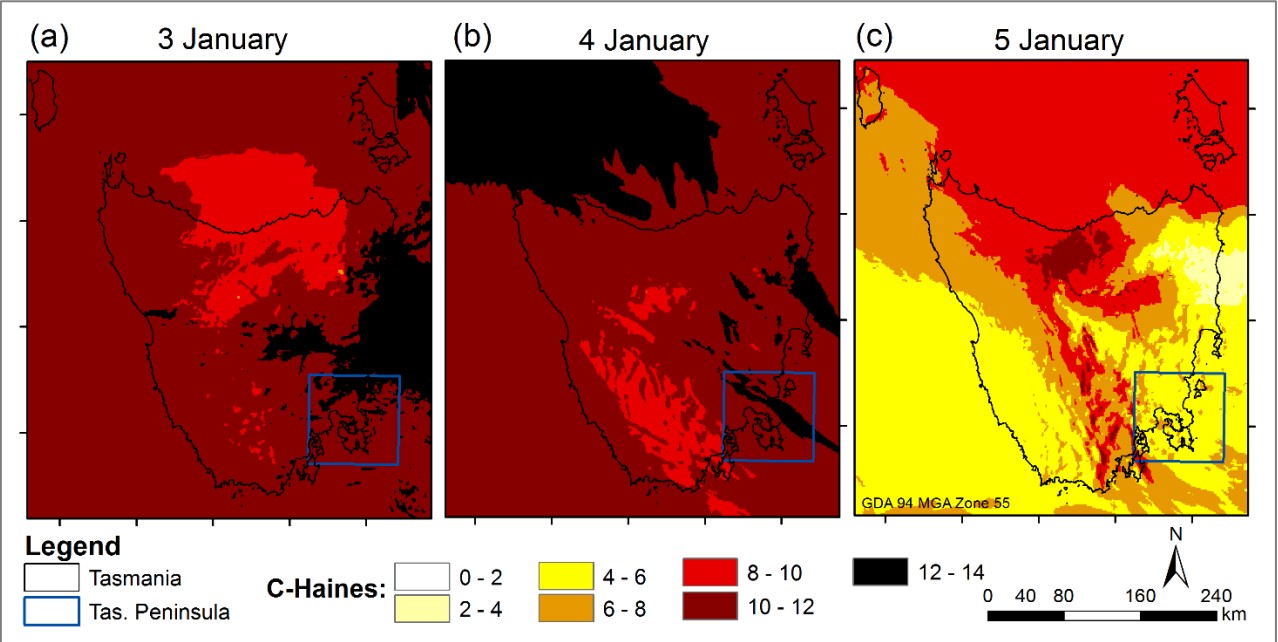

**Figure 4:** Spatio-temporal distribution of maximum C-Haines for Tasmania for 3-5 January 2013. The blue square indicates the location of the Tasman Peninsula.

## 3.2 Contextualising the Forcett-Dunalley pyroCb

### 3.2.1 Fire weather in large Tasmanian fires

The Forcett-Dunalley fire had amongst the highest levels of elevated fire weather (gridded FFDI and C-Haines of 68 and 11.5, respectively) of all the 77 large (>500 ha) Tasmanian fires that occurred between 2007-2016 (Fig. 5). Figure 5 shows the fire weather associated with all of these fires overlaid on the bivariate density distribution of FFDI and C-Haines for all days in the available record. The figure shows that despite being correlated, the probability of concurrence of elevated C-Haines and FFDI values across Tasmania is low. Further, the scatterplot suggests that high FFDI (>25) does not influence large fire occurrence as much as high C-Haines (>9), and that most large fire events (52 fires or 68%) occur within lower FFDI and C-Haines thresholds (15 and 7, respectively). Notably, the Forcett-Dunalley fire is an outlier and the only known fire to have produced a pyroCb.




**Figure 5:** Scatterplot of maximum daily FFDI and associated daily C-Haines (corresponding to the time of the highest FFDI) of each of the 77 large (>500 ha) wildfires overlaid on the density of all days in the period of the BARRA data. The density scale has been log-transformed for visualisation. The vertical and horizontal lines represent thresholds of elevated FFDI (25) and C-Haines (9) respectively. The Dunalley-Forcett fire is a clear outlier.


### 3.2.1 Spatio-temporal variability of elevated fire weather

During the fire season, eastern Tasmania and the Bass Strait Islands (Flinders and King Islands) are prone to combined elevated C-Haines and FFDI values (Fig. 6), particularly so for southeast Tasmania. Outside the fire season, only south-eastern Tasmania is exposed to the risk of elevated C-Haines and FFDI. There is no trend in elevated fire weather detectable in the

existing BARRA weather record across Tasmania (Fig. S6) but as longer reanalysis datasets become available, examination of longer trends may be possible.


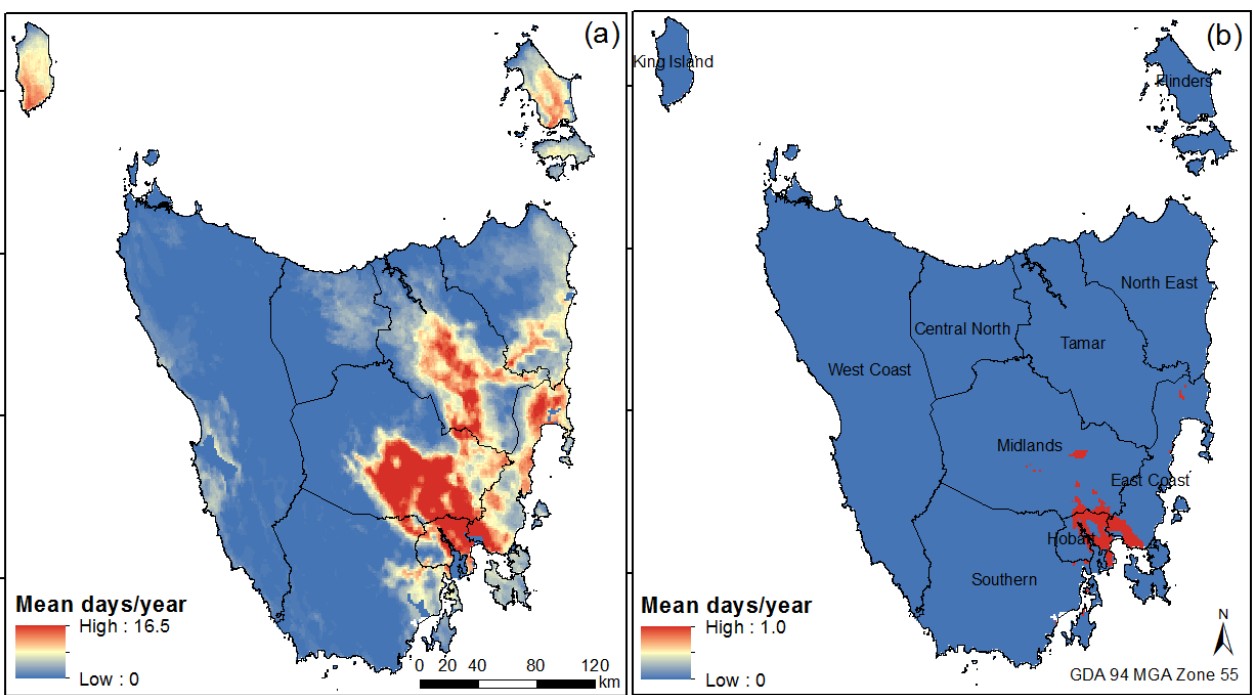

**Figure 6:** Spatial distribution of mean days per year with both elevated C-Haines (>9) and FFDI (>25) for, (a) the fire season (October-March) and (b) the non-fire season (April-September), intersected by operational fire management boundaries for Tasmania.

## 4 Discussion

We have described the evolution of the only known pyroCb to have occurred in Tasmania. Our study opportunistically used weather radar to track violent pyroconvection, which reached a peak height of 15 km in less than one hour during the afternoon of 4 January. We were able to associate this rapid escalation of the fire with a previously developed fire severity map and a digital terrain model of the fireground (Ndalila et al., 2018). Analysis of gridded weather data in Tasmania for the period 2007-2016 showed that this event is an outlier in elevated values of FFDI and C-Haines. Below we discuss these findings with respect to the known drivers of pyroCbs, and how it relates to previous pyroCb events in Australia and globally.

The Forcett-Dunalley firestorm developed in the late afternoon – a pattern similar to all reported Australian pyroCbs (Dowdy et al., 2017; Fromm et al., 2006; Peace et al., 2017), except the 2006 Grose Valley pyroCb in New South Wales (Fromm et al., 2012) and a second Waroona pyroCb event (Peace et al., 2017), which occurred in the late morning. For instance, the Kinglake fire, that was among the 2009 Victorian 'Black Saturday' fires – the most destructive of all known Australian fires – had extensive pyroCb activity in the late afternoon (17:00 LT) (Dowdy et al., 2017). PyroCbs are typically short-lived events that mature in less than an hour as was the case for the Forcett-Dunalley fire, although some can last for multiple hours such as the 2003 Canberra fires (Fromm et al., 2006) and the 2017 Pacific Northwest wildfires in north-western US and British Columbia



(Peterson et al., 2018). The Forcett-Dunalley pyroCb achieved a height of 15 km, which is similar to that reported in other Australian events: for example, the 2003 Canberra fires (14 km, Fromm et al., 2006), the 2006 Wollemi fire in New South Wales (14 km, Fromm et al., 2012), the 2009 Black Saturday fires (~13 km, Cruz et al., 2012), the 2016 Waroona fire in

Western Australia (14 km, Peace et al., 2017) and the recent 2019 Victorian fires (Mike Fromm, unpublished data). Further, two lightning strikes from the Forcett-Dunalley pyroCb were observed to the southeast of Dunalley, over the Tasman Sea. A majority of the Australian pyroCbs have been observed to generate lightning, which in some cases started satellite fires well ahead of the fire front (Peace et al., 2017). For instance, the Kinglake fire produced several lightning clusters, which ignited a new fire 100 km downwind of the fire (Dowdy et al., 2017). The 2003 Canberra pyroCb did not produce lightning, but is

notable for being the only confirmed case in Australia that produced a genuine tornado (peak diameter of 400-500 m) which led to extensive vegetation damage along its path (McRae et al., 2013).

The Forcett-Dunalley fire, and other known Australian pyroCbs, share characteristics of events in North America in terms of height (>12 km; Diner et al., 2004; Fromm et al., 2005; Dahlkotter et al., 2014), lightning activity (Johnson et al., 2014;

Rosenfeld et al., 2007), time of occurrence (Rosenfeld et al., 2007; Peterson et al., 2017), although some events have been reported to have lasted up to five hours (Peterson et al., 2018; Ansmann et al., 2018). North American studies have also highlighted the clear role that mid-tropospheric moisture plays in driving pyroCb development in the western US and Canada (Trentmann et al., 2006; Peterson et al., 2017), but this role does not appear as clear-cut in Australian pyroCbs. Indeed, a number of Australian pyroCb events exhibit a distinct lack of midlevel moisture in their associated atmospheric profiles (*e.g.*

the 2003 Canberra fire (Fromm et al., 2006) and the 2013 Wambelong fire (Wagga Wagga sounding for 13 January 2013 on http://weather.uwyo.edu/upperair/sounding.html). Further research is required to properly understand the potential influence of mid-tropospheric moisture in driving pyroCb development in Australia.

PyroCb development has been shown to be influenced by critical fire weather events. For instance, nocturnal foehn winds

occurred the night preceding pyroCb development in both of the Forcett-Dunalley and Grose Valley fire cases. These warm, dry winds disrupt the fuel moisture recovery phase that usually occurs overnight, thereby priming the landscape with drier fuels the following day (McRae et al., 2015). PyroCb development can also be influenced by the passage of troughs or wind changes such as cold fronts or sea-breezes (Peace et al., 2017; Mills and McCaw, 2010). Troughs provide a thermodynamic environment more favourable for moist convection in general, and the added lift from wind changes is thought to give pyroCb

formation a boost. Recent research (Tory et al., 2018) also highlights the potential role that a wind change may play in enhancing pyroCb development by lowering the plume condensation height, through entrainment of cooler and moister air. However, the Forcett-Dunalley pyroCb established itself well before the arrival of the wind change, as was the case with a number of other notable Australian pyroCbs such as the Grose Valley fire (McRae et al., 2015).




Extreme fire weather in eastern Tasmania reflects the combination of: (1) dominance of flammable vegetation, mainly composed of *Eucalyptus* forest and woodlands; (2) foehn-like winds (Grose et al., 2014; Fox-Hughes et al., 2014) and (3) pre-frontal troughs and cold fronts that cause elevated fire danger and rapid changes in fire spread (Grose et al., 2014; Fox-Hughes

et al., 2014; Bureau of Meteorology, 2013; Sharples et al., 2010; Cruz et al., 2012). Southeast Tasmania is particularly prone to foehn winds in pre-frontal environments, making the region vulnerable to wind direction changes that cause fire flanks to become fire fronts, thereby rapidly escalating the size of fires. Our analysis has shown that the Forcett-Dunalley fire was affected by these meteorological factors, having the highest levels of elevated fire weather of all the Tasmanian fires between 2007-2016 and was the only event to have produced a pyroCb. An important framework to understand the Forcett-Dunalley

fire is the analysis of fire weather conditions during 40 pyroCb events in south-eastern Australia conducted by Di Virgilio et al. (2019) that showed the environmental conditions conducive for pyroCb development are: extreme C-Haines conditions (10-13.7) and Very High to Catastrophic near-surface fire danger (FFDI 25-150) occurring over forested and rugged landscapes. The Forcett-Dunalley pyroCb event is consistent with these observations, occurring under elevated C-Haines (10-11), Severe-Extreme FFDI (60-75) on undulating/rugged terrain that supported long unburnt, dry *Eucalyptus* fuels. Di Virgilio

et al. (2019) suggest that extreme FFDI and C-Haines values may not lead to pyroCb development if there is a lack of deep flaming, for instance, due to elevated fuel moisture, low fuel loads or fire suppression activities. This suggests that the heavy fuel loads that resulted in a large area burnt at high severity in the Forcett-Dunalley fire (Ndalila et al., 2018) may have contributed to the development of the pyroCb.

Mainland Australia, and especially the south-eastern region, experiences a greater number of days of elevated values of FFDI and C-Haines than Tasmania (Dowdy and Pepler, 2018). The FFDI/C-Haines distribution in that region shows pyroCbs occurring over a wide range of FFDI but under more limited extreme values of C-Haines (Di Virgilio et al., 2019). In Tasmania, however, we found a fire weather environment conducive for pyroCb occurrence is likely to have high concurrent values of C-Haines and FFDI (Figure 5), a conjunction that occurs rarely, although this may change with the projected warming climate

in Australia (Dowdy, 2018; Di Virgilio et al., 2019). It is worth noting that the FFDI/C-Haines distribution for Tasmania reveals that high FFDI (>25) does not influence large fire occurrence as much as high C-Haines (>9), consistent with findings of Di Virgilio et al. (2019). Climate modelling in Australia suggests that climate change may lead to an increased risk of strong pyroconvection, particularly in the spring months in south-eastern Australia (Di Virgilio et al., 2019; Dowdy and Pepler, 2018). Within Tasmania, climate models point to increased FFDI and longer fire seasons (Fox-Hughes et al., 2014; Grose et al., 2014),

hence it is likely that the risk of pyroCb will increase, with eastern and south-eastern Tasmania particularly vulnerable, given the concentration of high C-Haines values identified in this study.





Turbulence associated with winds and terrain has been suggested as an important factor contributing to pyroCb formation via amplification of fire through processes such as mass spotting, topographic channelling of winds and vorticity-driven lateral spread (VLS) (Sharples et al., 2012). Other studies have linked abrupt increases in plume height with VLS; these include the 2003 Canberra fires (Sharples et al., 2012) and the 2006 Grose Valley fire (McRae et al., 2015). Our observation of rapid lateral expansion of the plume just before the pyroCb attained its maximum height (Fig. S4) is suggestive of a VLS occurrence

(McRae, 2010). Mapping of the Forcett-Dunalley fire by Ndalila et al. (2018) showed that areas subjected to the highest fire intensities broadly aligned with terrain prone to VLS. The VLS mechanism may also explain the ember storm that impacted the coastal township of Dunalley situated in the lee of the low hills. It must be acknowledged that the role of downdrafts and mass spotting underneath the plume during the period of extreme fire behaviour is hard to infer without additional data sources on fire behaviour (such as infra-red/multispectral linescans) and high-resolution coupled fire-atmosphere modelling (Peace et

al., 2015).

This study hinges on the application of weather radars to track the evolution of a pyroCb. It is worth noting that weather radars are not perfectly suited for all fires because of their limited geographic range (relative to satellite observations) and their inability to detect microscale cloud particles (<100 μm). Nonetheless, radars remain a reliable data source that can provide

near-real time monitoring of strong pyroconvection, as evidenced by previous pyroCb studies in Australia and globally (Rosenfeld et al., 2007; Fromm et al., 2012; Dowdy et al., 2017; Lareau and Clements, 2016; Lareau et al., 2018; Peace et al., 2017). A feature of our study was linking plume evolution to fire severity mapping - an approach that has received limited attention. Duff et al. (2018) conducted one of the few studies, using statistical models to link the radar-detected plume volume to fire growth, and found that radar return volume (above a threshold of 10 dBZ) was a robust predictor of fire-area change.

These results appear consistent with the present findings, particularly with the correlation between rapid plume development and the lateral growth of the fire.

The present study is the first in Tasmania that has utilized geographically comprehensive fire weather information both at the near-surface and the lower atmosphere to determine the spatio-temporal variation of elevated fire weather in Tasmania. One

opportunity for further research is to examine long-term trends in elevated fire weather as longer weather reanalysis datasets for Tasmania (1990-2019) become available. The influence of climate change on spatial and seasonal dynamics of C-Haines is particularly important for comparison with projected changes in south-eastern Australia conducted by Di Virgilio et al. (2019).

Our analysis contributes to improving the prediction of extreme fire behaviour (Tory and Thurston, 2015). Indeed, in this context it is interesting to know if the Forcett-Dunalley pyroCb could have been predicted with existing information. We suggest it is conceivable that the Forcett-Dunalley pyroCb blow-up between 15:30 and 16:30 LT on 4 January 2013 could have been predicted potentially 12-18 hours in advance. Further, the Phoenix fire behaviour model had predicted in the evening of





3 January that the fire would reach Dunalley at approximately 15:00 on 4 January (Bureau of Meteorology, 2013), which is
around the time the fire reached Dunalley (15:24-15:48) following the predicted path, confirmed by plume/pyroCb dynamics
(Fig. S3) and witness reports. The conditions surrounding the pyroCb event are entirely consistent with those highlighted in
the Blow-Up Fire Outlook (BUFO) model (McRae and Sharples, 2013, 2014). In essence, the BUFO model assesses the
likelihood of a fire exhibiting deep flaming in an atmospheric environment conducive to rapid plume growth. Retrospective
application of the BUFO model to the Forcett-Dunalley case yields the BUFO pathway summarised in Fig. S7. It is initiated
by the presence of an uncontrolled fire in elevated fire weather, combined with wind speed of >25 km h$^{-1}$ over rugged forested
landscapes, with dead fuel moisture below 5%, in a VLS-prone land form and under extreme values of atmospheric instability
and dryness (C-Haines ≥10).

## 5 Conclusion

This study has provided an analysis of pyroCb dynamics and fire weather during an extreme fire event in Tasmania on 4
January 2013. We have shown that the pyroCb was associated with elevated fire weather conditions, and strong interactions
between weather, terrain and the fire itself, which caused dynamic fire behaviour and the near-destruction of Dunalley
township. We have discussed the known drivers of pyroCbs, and how the Forcett-Dunalley pyroCb relates to previous pyroCb
events in Australia and globally. An analysis of fire weather in previous large wildfires in Tasmania between 2007-2016
suggests that the Forcett-Dunalley fire experienced among the highest levels of elevated fire weather of all the large fires in
Tasmania, and was the only event to have produced a pyroCb to date. A spatio-temporal analysis of fire weather in Tasmania
shows eastern (particularly south-eastern) Tasmania is subject to more days of elevated fire weather than the west, highlighting
the vulnerability of this region to extreme fire events. This information is crucial for fire weather forecasting and fire
management and planning.

**Data availability:**

The gridded weather reanalysis data for Tasmania (BARRA-TA) for the period 2007-2016 are provided upon registration with
the reanalysis helpdesk at Bureau of Meteorology (BoM) via the link: http://www.bom.gov.au/research/projects/reanalysis/.
Weather radar data (in Odim HDF5 format) are publicly available for non-commercial use via the Open Radar Data program
at BoM through http://dapds00.nci.org.au/thredds/catalog/rq0/catalog.html. The upper air (radiosonde) data is also available
from BoM as two daily observations (at 9:00 and 21:00 Local Standard Time). Global radiosonde data can be obtained from
the University of Wyoming interactive mapping portal (http://weather.uwyo.edu/upperair/sounding.html). The fire history
dataset for Tasmania as well as fire management area (FMA) boundary data for Tasmania are available from unpublished
records held by Tasmania Fire Service; while the 33 m Digital Elevation Model is available from the Tasmanian Department
of Primary Industries, Parks, Water and Environment (DPIPWE).





**Author contribution:** M.N.N., G.J.W., P.FH, J.S. and D.M.J.S.B. designed the experiments; P.FH, provided meteorological datasets; M.N.N. analysed the data, while G.J.W developed the R code for extraction and conversion of the gridded weather reanalysis (BARRA-TA) product and the weather radar dataset. M.N.N. prepared the manuscript with contributions from all co-authors.

**Competing interests:** The authors declare that they have no conflict of interest.

**Acknowledgement:**

This research was funded by the ARC Linkage grant (LLP130100146), and the Bushfire and Natural Hazard Cooperative Research Centre (BNHCRC) through a top-up scholarship to the main author. We thank Tom Remenyi of the University of Tasmania and the reanalysis team at the Bureau of Meteorology (BoM) for facilitating online access to the BARRA-TA product, Joshua Soderholm of BoM for providing additional weather radar datasets for this project, and Kevin Tory of BoM for his advice on some meteorological aspects of the study. We are also grateful to Rick McRae of the ACT Emergency Services Agency for assistance with the BUFO analysis. The support of the University of Tasmania, and the end-user support from Sam Ferguson of Tasmania Fire Service, through BNHCRC funding is gratefully acknowledged.

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
