# Peer review of "Evolution of a pyrocumulonimbus event associated with an extreme wildfire in Tasmania, Australia"

_Natural Hazards and Earth System Sciences, 2019_

## Referee Comment (RC1) · Paulo Fernandes (Referee) · 11 Jan 2020

The manuscript subject is the analysis of a pyroCb event in Tasmania, the first on record, that occurred in the Forcett-Dunalley on 2013. Its formation and evolution is related to fire danger and the C-Haines index, as well as with fire severity. I found the work solid and writing effective. The only issue of note is the role attributed to VLS (vorticity-driven lateral spread) by the authors. From the information and figures provided I really don't see enough empirical evidence for it, so I think the text could be more cautious. See below specificcomments addressing mostly minor concerns.

L11. I am not sure whether pyroCb development can be called a fire characteristic.

L31. 'energetically intense". Fire intensity is energy by definition. Maybe rephrase to

'high-intensity' or similar.

L32. Shouldn't it be 'conducive to'?

L35. Check ';', it's breaking the sentence flow.

L35-36. Temperature is essentially expressed through fuel dryness. So remove the former or remove the later and add relative humidity.

L37. This is a narrow definition of an extreme fire, as deep flaming and atmosphere coupling are often absent. Better to rephrase to make more explicit what the authors consider extreme, i.e. a particular type or range of extreme fire behaviour.

L46. 'Near the pyroCb' suggests it is near the surface.

L64. More accurately, the C-Haines index indicates the potential for large fire development.

L65. From 0 to 13?

L190. 'These thresholds were chosen to define an elevated fire weather day (95th percentile)' is confusing. Maybe 'to correspond' instead of 'to define'?

L218. The figure would benefit from the inclusion of wind direction. Did it change noticeably during the fire duration and is there information about wind direction at different heights? This can be important in relation to the assumed/inferred VLS subsequently in the text. Changes in wind direction impact the fire plume and can change spotting patterns hence modifying fire growth rate and direction.

L239 (Figure 3). So, fire severity distribution is for three periods? Wouldn't it be possible to partition those more? This would require fire growth isochrones or perhaps take advantage of the information provided by Jon Marsden-Smedley report? Also, to supplement fire severity and context, the text could indicate (possibly in the discussion) rates of spread of the wildfire for those periods, as they are available (at least in part, did not check) in said report.

L255. I know it is given in another paper, but readers would benefit from additional information about fire severity in the Methods section.

L256. See comment regarding L239.

L291. 500 ha is not that high as a large fire size threshold. I would appreciate an enhancement of this figure where the green dots were attributed different colours corresponding to different fire size classes.

L327. 'satellite fires'?

L425. And yet, Phoenix does not incorporate fire-atmosphere relationships, right?

L436. The interactions with terrain were suggested, rather than shown. From what I see in the supplementary material the VLS areas were quite small and fragmented in the landscape.

Enjoyed reading the manuscript,

Paulo Fernandes

---

## Short Comment (SC1) · 22 Jan 2020

This article by Ndalida et al. presents the concomitant analysis of a wildfire plume, which produced a pyroCb, and surface fire behaviour. I quite enjoyed reading the paper, and I am quite familiar with the area where the fire took place. My research interests include the use of weather radar to monitor fire plumes, therefore I felt it useful to provide some comments that might help improve the paper.

Overall, I feel that the radar data, possibly making half of the paper content (the other half being fire behaviour observations) are well under-utilised by the authors. For instance, I am surprised that the authors did not show and discuss radar observations of the Doppler velocity? We have shown in Terrasson et al. (2019) that these can provide

great insights into the dynamics of the PyroCb, including convection, shear, vortices etc. Its derivative, i.e. the spectrum width as we showed in Terrasson et al. (2019) and McCarthy et al. (2017; 2018) can also give great insights. Here, the radar data is only presented as a time-series of integrated variables in one figure of the article, which gives limited information. The Figure S3 in the supplementary material shows only cross sections of the Reflectivity, without colour scale, which is really limited. Recent work by McCarthy et al. (2019) and number of papers from Lareau and Clements show that radar observations can be utilised to draw multiple quantitative and qualitative information: this richness must be utilised.

Specific comments below:

1. Title: "...Pyrocumulonimbus-driven...." We know of the feedback loop between surface fire behaviour and pyroCb, but it reads as if the fire was influenced by the PyroCb and not the other way around. What do you mean by "Extreme PyroCb", how does it differ from a standard PyroCb? The use of more scientific rather than emotive language should be preferred.

2. L34 and other occurrences in the article. Shouldn't references listed in brackets be in chronological order?

3. L36: this seems somehow restrictive, as not only atmospheric instability but wind shear and mesoscale conditions can drive fire behaviour.

4. L44; Chronological order

5. Melnikov et al. (2008) does not present observations of PyroCb so the reference should not be cited here. Authors should also cite Terrasson et al. (2009) in which we report detailed radar observations of a PyroCb in NSW.

Terrasson, A., McCarthy, N., Dowdy, A., Richter, H., McGowan, H., & Guyot, A. ( 2019). Weather radar insights into the turbulent dynamics of a wildfire‐triggered supercell thunderstorm. Journal of Geophysical Research: Atmospheres, 124, 8645– 8658.

[Figure]

https://doi.org/10.1029/2018JD029986

6. L57 and 58: This sentence would probably benefit from the use of a more scientific description of the radar capabilities, e.g. ..."they are "tuned" to identify" is actually incorrect. The frequency of the radar will determine which particles the radar is likely to receive a backscatter from. Post-processing analysis can be improved possibly, but it won't modify the frequency of the radar, which is a fixed hardware choice. More details on the scatterers as observed by the radar and reference to pyrometeors as described by McCarthy et al. (2019) would be appropriate here.

McCarthy, N. F., Guyot, A., Protat, A., Dowdy, A., & McGowan, H. ( 2019). Tracking Pyrometeors with Meteorological Radar Using Unsupervised Machine Learning. Geophysical Research Letters, 46. https://doi.org/10.1029/2019GL084305

7. L69: the authors might want to discuss here the V-shape profile as described by Peterson et al.?

8. L71: it would be useful to give here the frequency of the radar (C-band) and state that this is a Doppler radar. The reader might also want to know if this is an operational radar (thus with a given scan strategy) and by whom it is operated (BoM).

9. L83: space between "600" and "m"

10. Can you give here the time at which the pyroCb started to form?

11. Figure 1(d), the location of the weather radar is hard to see... could you possibly use another colour or/and larger font?

12. L108: This is likely to be case but how did McRae validate this? What do you mean by "violent convection"? i.e. did you compute vertical velocities?

13. L111: there is no consistency with the spelling of "Mt" and "Mt." in the paper. I think it should be "Mt"?

14. L119: Why is that threshold of 11 dBZ being used? The paper would benefit here

from more explanation on the processing of radar data (clutter removal, attenuation correction if any etc.)

15. L119 and 120: "in this paper" repeated twice

16. L162 and elsewhere: it might be good to use the terminology "air temperature" instead of "temperature".

17. L165: section title seems wrong here

18. Figure2: Wind direction? This is an important factor for fire behaviour and could help discuss the VLS aspects presented by the authors as one of the main mechanism for the increase in fire intensity? I personally prefer multiple subpanels as there is a bit of clutter on that figure with all the variables.

19. L224: "1km" above sea level?

20. Figure 3: "Smoke on subpanel (a)" is not technically correct; rather "smoke plume" should be used. That is because radar at C-Band can't get good returns from smoke as such, as the authors correctly mentioned earlier in the draft; see McCarthy et al. (2018).

21. L249 How is plume length defined?

22. Figure 5, nice

23. Maybe recall for readers unfamiliar with Tasmanian seasons when that is?

24. L315 The authors should cite Terrasson et al. here

25. L322 325 The authors should cite Terrasson et al. here as well

26. L338. The authors should discuss the findings of Terrasson et al. – the change of moisture in the lower and upper levels as brought by the cold front on the development of PyroCb and fire behaviour.

27. L348: The authors should cite Terrasson et al. here

28. L392, 394: The effect of the potential VLS on plume development is too speculative. Overall, I think that there is no justification based on the observations as shown in the current paper, for any effect of VLS.

29. The authors should cite Terrasson et al. here

30. In Terrasson et al. (2019) we did exactly that, e.g. linking fire behaviour to plume development. The authors could also cite McCarthy et al. (2018) where fire behaviour is studied against plume development for a PyroCu in Victoria (Mt Bolton fire).

McCarthy, N., H. McGowan, A. Guyot, and A. Dowdy, 2018: Mobile X-Pol Radar: A New Tool for Investigating Pyroconvection and Associated Wildfire Meteorology. Bull. Amer. Meteor. Soc., 99, 1177–1195, https://doi.org/10.1175/BAMS-D-16-0118.1

31. Fig.S3: "the orange colour represents the most intense parts of the PyroCb": it would be appropriate to use a more scientific and precise language, i.e. provide the values or range of values for the equivalent reflectivity in dBZ (colour bar), and refer to the text of the article to describe how these have to be interpreted.

Thanks and again I enjoyed reading your study. Kind regards,

A. Guyot

---

## Referee Comment (RC2) · Anonymous Referee #2 · 24 Jan 2020

General comments

The manuscript entitled "Evolution of an extreme Pyrocumulonimbus-driven wildfire event in Tasmania, Australia" provides an analysis of the extreme fire event in Tasmania of 4 January 2013. The temporal evolution of the PyroCb is assessed using weather radar data and is then analyzed together with the McArthur Forest Fire Danger Index (FFDI), the Continuous Haines Index (C-Haines) and fire severity maps. Overall, the presentation quality is good, the results are well described and discussed, and the manuscript covers a relevant topic that has been gaining increasing attention over the last years. As such I believe that a revised version of the manuscript will present a good contribution to the fire community.

My main concern is regarding section 3.2.1 analysis of large wildfires (>500 ha) in

the context of FFDI and C-Haines. Reasoning only about the number of large fires in the FFDI/C-Haines domain (Figure 5) can be misleading since, for example, when comparing the number of large fires for FFDI >25 or <25, as the number of days with FFDI < 25 is much higher, the interpretation of the figure may be biased. I would suggest plotting the smaller fires in other colour and computing for each of the 4 regions (delimited by the dashed lines) in Figure 5, the fraction of fires exceeding 500 ha. Reasoning about the fraction of large fires may be more insightful for the discussion.

Specific comments

L65: Is the range 0-13 correct? In Figure 4 the last class has a range of 12-14 and L367 refers values up to 13.7.

L165: 2.3? Also, a few sentences explaining the intuition of the VLS could be helpful.

Figure 3: What are the units for Max.height?

L255: What is the actual proportion for the 3 groups? It's hard to tell just from the image.

L265: The 0.5 correlation is for 1-day lag?

L282: It may be relevant to also comment on how extreme are these values regarding all days, for example, what percentile they correspond, 95, 99?

L363: Is this true for all fires or for all large fires (>500 ha)?

---

## Referee Comment (RC3) · Anonymous Referee #3 · 13 Feb 2020

This paper provides an analysis of a significant pyroCb event in Tasmania (January 2013). The evolution of the plume is assessed using weather radar. Analysis of meteorology is largely based on the lower troposphere using the C-Haines index and FFDI. Fire severity maps are also included. The manuscript is generally well-organized and easy to read. Results are relevant to several research communities. The pyroCb phenomenon has been gaining significant attention in recent years. However, a few aspects of the paper require some clarification prior to publication. Please see my comments below.

Abstract line 18: "highly unstable atmosphere" do you mean "highly unstable lower troposphere" since you are referencing C-Haines?

Lines 65-70: lapse rate in lower-troposphere, mid-levels, or both?

[Figure]

Line 139: "10 hPa"….what are these high altitude data used for? C-Haines is only based on the lower troposphere?

Line 181: is there a reference for "typically happens at or near the start of fires"?

Line 195: please replace "was always higher than" with a specific range of values.

Section 3.1: In addition to surface fire weather, pyroCb are driven by the full 3-D structure of the troposphere. This discussion can be improved. The supplement provides thermodynamic profiles. Can you quantify the LCL and highlight the pressure levels that are used to calculate C-Haines? Was there any CAPE in these profiles? Please also provide a couple sentences describing the synoptic weather pattern that set the stage for this event.

Section 3.1.2: Please show a basic radar image/map at the peak of the pyroCb event for reader orientation.

Line 233: please briefly describe the significance of the 850-500 lapse rate in the main text.

Line 265: "good correlation of 0.5"….what is good?

Line 269: "wind change at around 00:00"…is this a cold front? Some description is likely needed?

Section 4: Did you examine additional radar products for this case, such as velocity data? You might be able to get more information on the dynamics of the updraft and even particle characteristics. Radar data are one of the more underutilized tools in pyroCb research.

Lines 335-345: The soundings in the supplement do show the inverted-V structure typical of high-based convection, with some mid-level moisture in the profile at 1500. You can calculate total precipitable water to confirm. I believe that Peterson et al. provide more info on this.

Supplement: Please annotate the time of the pyroCb on the relevant figures.

Supplement: Define/explain "BARRA pseudo-soundings"

It would be nice to include a surface and upper level synoptic weather map that coincides with the soundings in Figure S1. This will provide some context on the large-scale meteorology driving this event.

Are the times listed local time or UTC?

---

## Author Comment (AC1) · 21 Mar 2020

RC: This article by Ndalila et al. presents the concomitant analysis of a wildfire plume, which produced a pyroCb, and surface fire behaviour. I quite enjoyed reading the paper, and I am quite familiar with the area where the fire took place. My research interests include the use of weather radar to monitor fire plumes, therefore I felt it useful to provide some comments that might help improve the paper.

Overall, I feel that the radar data, possibly making half of the paper content (the other half being fire behaviour observations) are well under-utilised by the authors. For instance, I am surprised that the authors did not show and discuss radar observations of the Doppler velocity? We have shown in Terrasson et al. (2019) that these can provide

great insights into the dynamics of the PyroCb, including convection, shear, vortices etc. Its derivative, i.e. the spectrum width as we showed in Terrasson et al. (2019) and McCarthy et al. (2017; 2018) can also give great insights. Here, the radar data is only presented as a time-series of integrated variables in one figure of the article, which gives limited information. The Figure S3 in the supplementary material shows only cross sections of the Reflectivity, without colour scale, which is really limited. Recent work by McCarthy et al. (2019) and number of papers from Lareau and Clements show that radar observations can be utilised to draw multiple quantitative and qualitative information: this richness must be utilised.

AC: We greatly appreciate the reviewer's comments about making further use of the radar data. However, we feel that including all of the work suggested by the reviewer would go well beyond the intended scope of the manuscript, which was to provide a qualitative description of the evolution of the Forcett-Dunalley fire and pyroCb, and their relation to coincident surface weather and lower atmospheric conditions. Indeed, we feel that it would be more appropriate to pursue the work suggested by the reviewer in a separate and more detailed quantitative analysis, drawing upon the work of Terrasson et al. (2019) and McCarthy et al. (2019). We have amended the discussion to specifically note this as a natural extension of the qualitative analysis presented in the manuscript.

Additionally, an aspect of the Sir Ivan fire in Terrasson et al. (2019) was that the pyroCb was a supercell, which is a more organised type of thunderstorm, with more to investigate in the radar data. There's no evidence of that in Dunalley, but this might make a similar analysis more interesting as a complement to that study.

The section in the discussion now reads: "... radars remain a reliable data source that can provide near-real time monitoring of strong pyroconvection, as evidenced by previous pyroCb studies in Australia and globally (Rosenfeld et al., 2007; Fromm et al., 2012; Lareau and Clements, 2016; Dowdy et al., 2017; Peace et al., 2017; Lareau et al., 2018; Terrasson et al., 2019). This study did not analyse radial velocity from

the Doppler radar; therefore future research on the Forcett-Dunalley fire and other fires should consider using that information to provide a more quantitative analysis of the thunderstorm, drawing upon previous work in Australia (McCarthy et al., 2019; Terrasson et al., 2019). A feature of our study was linking plume evolution to fire severity mapping - an approach that has received limited attention".

RC: Specific comments below: RC: 1. Title: " . . .Pyrocumulonimbus-driven . . .." We know of the feedback loop between surface fire behaviour and pyroCb, but it reads as if the fire was influenced by the PyroCb and not the other way around. What do you mean by "Extreme PyroCb", how does it differ from a standard PyroCb? The use of more scientific rather than emotive language should be preferred.

AC: Noted. Actually, extreme in this case describes the wildfire event, not the pyroCb. Regarding the statement that the title supposes that the pyroCb caused the extreme fire, that is also true, although we meant, as you correctly put, that the extreme fire caused the pyroCb.

A new title now reads: "Evolution of a pyrocumulonimbus event associated with an extreme wildfire in Tasmania, Australia".

RC: 2. L34 and other occurrences in the article. Shouldn't references listed in brackets be in chronological order?

AC: Noted. All the references are now in chronological order.

RC: 3. L36: this seems somehow restrictive, as not only atmospheric instability but wind shear and mesoscale conditions can drive fire behaviour.

AC: We have rephrased the sentence to read: "While fire weather is most often understood as a surface phenomenon (for example, through surface temperature, wind speed and relative humidity), atmospheric processes such as instability, wind shear and mesoscale conditions can also drive extreme fire development".

RC: 4. L44; Chronological order

AC: Noted. The references are now in chronological order.

RC: 5. Melnikov et al. (2008) does not present observations of PyroCb so the reference should not be cited here. Authors should also cite Terrasson et al. (2009) in which we report detailed radar observations of a PyroCb in NSW. Terrasson, A., McCarthy, N., Dowdy, A., Richter, H., McGowan, H., & Guyot, A. ( 2019). Weather radar insights into the turbulent dynamics of a wildfire- triggered supercell thunderstorm. Journal of Geophysical Research: Atmospheres, 124, 8645–8658.https://doi.org/10.1029/2018JD029986

AC: We have replaced Melnikov et al. (2008) reference with Terrasson et al. (2009).

RC: 6. L57 and 58: This sentence would probably benefit from the use of a more scientific description of the radar capabilities, e.g. . . ."they are "tuned" to identify" is actually incorrect. The frequency of the radar will determine which particles the radar is likely to receive a backscatter from. Post-processing analysis can be improved possibly, but it won't modify the frequency of the radar, which is a fixed hardware choice. More details on the scatterers as observed by the radar and reference to pyrometeors as described by McCarthy et al. (2019) would be appropriate here.

McCarthy, N. F., Guyot, A., Protat, A., Dowdy, A., & McGowan, H. ( 2019). Tracking Pyrometeors with Meteorological Radar Using Unsupervised Machine Learning. Geophysical Research Letters, 46. https://doi.org/10.1029/2019GL084305

AC: We respectfully disagree that the statement is incorrect. We however agree with you that the frequency of radar determines what kind of particles it receives backscatter from. In line 104 of our original submission, we noted that weather radars do not detect smoke components smaller than 100 $\mu$m. Therefore, we believe that the above statement is correct.

For clarity, we have replaced 'tuned to' with 'sensitive' as follows: "Weather radars . . ... do not accurately detect the exact extent of entrained gaseous and fine particulate

emissions because they are sensitive to larger particles such as rain and ice crystals, and can therefore fortuitously detect pyrometeors such as ash, scorched debris and embers (McCarthy et al., 2019). We checked the reference you provided, whose authors agree with us by suggesting that there is "increasing evidence that weather radar does not have the sensitivity to detect aerosol-sized smoke targets. Instead it detects ash and larger debris, characterized as generally horizontal, plate-like targets with high (low) ZDR (hv)".

Additionally, the below reference states: "The consensus with such polarimetric radar results is now that the scatterers within fire plumes are dominantly ash". The article goes on to list all scatterers such as "coarse mode particle emissions, ash, firebrands, and extinguished and scorched debris."

McCarthy, N., Guyot, A., Dowdy, A., and McGowan, H.: Wildfire and Weather Radar: A Review, J. Geophys. Res. Atmos., 124, 266-286, doi:10.1029/2018jd029285, 2019.

RC: 7. L69: the authors might want to discuss here the V-shape profile as described by Peterson et al.?

AC: We have now added the following section: "The role of tropospheric temperature and moisture in pyroCb dynamics is exemplified in the inverted-V thermodynamic profile (Peterson et al., 2017). The profile shows a dry and warm near-surface environment in which temperature decreases dry adiabatically with altitude to the top of the mixed layer ($\sim$ 3 km) where relative humidity is higher. Altitudes immediately above the mixed layer are usually drier, and this dry air can mix to the surface in strong convective downdrafts, increasing surface fire behaviour (McRae et al., 2015). Further, higher mid-troposphere moisture can also interact with weaker wind shear and high temperature lapse rate to produce strong updrafts (Peterson et al., 2017)".

RC: 8. L71: it would be useful to give here the frequency of the radar (C-band) and state that this is a Doppler radar. The reader might also want to know if this is an operational radar (thus with a given scan strategy) and by whom it is operated (BoM).

AC: We have now included the word 'Doppler' in the sentence. The other suggested information has been provided in the description of the weather radar in section 2.2.1 (in the Methods section).

RC: 9. L83: space between "600" and "m"

AC: It has now been corrected.

RC: 10. Can you give here the time at which the pyroCb started to form?

AC: It is difficult to know the exact time, but likely between 15:24 and 15:30. A malfunction in radar between 15:30 and 15:36 resulted in plume information to be only available at the lowest elevation angle of the radar scan, and was therefore not useful.

The sentence now reads: "On 4 January, southeast Tasmania recorded dangerous fire weather conditions, resulting in a large uncontrollable fire that led to the development of a pyrocumulonimbus (Fig. 1Sb) from around 15:24 and caused the near-complete destruction of Dunalley township (Fig. 1d)".

RC: 11. Figure 1(d), the location of the weather radar is hard to see . . . could you possibly use another colour or/and larger font?

AC: The colour of the symbol has been changed and the size increased. See the figure attachment for details.

RC: 12. L108: This is likely to be case but how did McRae validate this? What do you mean by "violent convection"? i.e. did you compute vertical velocities?

AC: We did not compute vertical velocities, but the occurrence of the pyroCb (that rapidly rose from 9 km at 15:24 to 15 km at 15:48) is evidence of violent convection. The term 'violent convection' is used in the NWCG Glossary of Wildland Fire (dating back to 1949, most recent version is 2018) and has been adopted in the 2012 AFAC Bushfire Glossary. It entails a combination of strong (i.e. enough uplift to carry large particles that show up in radar data), highly turbulent and deep convection.

NWCG Glossary: https://www.nwcg.gov/glossary/a-z AFAC Glossary: https://www.afac.com.au/docs/default-source/doctrine/bushfire-terminology.pdf

For clarity, we have, in the introduction defined 'violent convection as 'strong, highly turbulent and deep convection', and maintained the term 'violent convection' thereafter in the manuscript.

RC: 13. L111: there is no consistency with the spelling of "Mt" and "Mt." in the paper. I think it should be "Mt"?

AC: Noted. They have been revised to "Mt" throughout the manuscript.

RC: 14. L119: Why is that threshold of 11 dBZ being used? The paper would benefit here from more explanation on the processing of radar data (clutter removal, attenuation correction if any etc.) We used the 11 dBZ threshold as it clearly demarcated the boundary of the plume better than lower values. We did not do detailed pre-processing such as what you have suggested. There is a degree of pre-processing of radar images prior to operational use (e.g. to remove clutter and permanent echoes), but we didn't further process the data. We contend that the effect of not conducting further processing would be negligible.

RC: 15. L119 and 120: "in this paper" repeated twice

AC: This has now been revised.

RC: 16. L162 and elsewhere: it might be good to use the terminology "air temperature" instead of "temperature".

AC: Noted. These have been revised.

RC: 17. L165: section title seems wrong here.

AC: The title has now been replaced by 'Vorticity-driven Lateral fire spread'.

RC: 18. Figure2: Wind direction? This is an important factor for fire behaviour and

could help discuss the VLS aspects presented by the authors as one of the main mechanism for the increase in fire intensity? I personally prefer multiple subpanels as there is a bit of clutter on that figure with all the variables.

AC: Wind direction has now been included, and plots separated into two: one for wind variables, the other one for the other weather variables. See figure supplement for details.

RC: 19. L224: "1km" above sea level?

AC: Yes. It now reads: "plume height from radar scans gradually increased from around 1 km above sea level at 13:00 to 8 km at around 15:00 and rapidly rose to the maximum injection height of 15 km. …."

RC: 20. Figure 3: "Smoke on subpanel (a)" is not technically correct; rather "smoke plume" should be used. That is because radar at C-Band can't get good returns from smoke as such, as the authors correctly mentioned earlier in the draft; see McCarthy et al. (2018).

AC: The legend in the subpanel (a) has now been corrected to reflect that it is a smoke plume. See the figure attachment for details.

RC: 21. L249 How is plume length defined?

AC: We have included the description as follows: "Plume length is the horizontal distance between the origin of the plume and the farthest extent of the plume".

RC: 22. Figure 5, nice.

AC: Thanks. However, other reviewers have suggested changes regarding stratifying the fires into different fire size classes, to which we have slightly modified. See figure supplement.

RC: 23. Maybe recall for readers unfamiliar with Tasmanian seasons when that is?

AC: The months in the fire season have now been included in the text. That information (including for non-fire season) is repeated in Fig. 6 captioning.

RC: 24. L315 The authors should cite Terrasson et al. here

AC: The article has now been cited.

RC: 25. L322 325 The authors should cite Terrasson et al. here as well

AC: The article has now been cited.

RC: 26. L338. The authors should discuss the findings of Terrasson et al. – the change of moisture in the lower and upper levels as brought by the cold front on the development of PyroCb and fire behaviour.

AC: It has now been included as follows: "... a number of Australian pyroCb events exhibit a distinct lack of midlevel moisture in their associated atmospheric profiles (e.g. the 2003 Canberra fire (Fromm et al., 2006) and the 2013 Wambelong fire (Wagga Wagga sounding for 13 January 2013 on http://weather.uwyo.edu/upperair/sounding.html). However, in this study, the mid- to upper-level moisture was higher during the time preceding pyroCb formation on 4 January (Fig. S2a), with a total precipitable water of 23 mm indicating a moist lower atmosphere (Webb and Fox-Hughes, 2015). Terrasson et al. (2019) report on the effect of a change in moisture between the low- and upper-levels (brought by a cold front) on the development of a pyroCb and enhancement of fire behaviour in the Sir Ivan fire in eastern Australia. Further research is required to properly understand the potential influence of mid-tropospheric moisture in driving pyroCb development in Australia".

RC: 27. L348: The authors should cite Terrasson et al. here

AC: The article has now been cited.

RC: 28. L392, 394: The effect of the potential VLS on plume development is too speculative. Overall, I think that there is no justification based on the observations as

shown in the current paper, for any effect of VLS.

AC: We agree that the VLS-prone areas seem small, and their influence on pyroCb occurrence likely negligible based on these results, but we also caution that VLS likely occurred, but the effect may have been undermined by the limitations of the input data (DEM and wind direction layer).

We have now rephrased the sentence in both the results and discussion sections to read: In section 3.1.3 (Results): "Evidence of an effect of Vorticity-driven Lateral Spread on the fire behaviour was not strong. Analysis of the precursor terrain conditions only revealed small patches of VLS-prone areas near Dunalley township (Fig. S4). However, we are not able to rule out VLS occurrence on parts of the terrain that were not resolved by the DEM, and which may have played a part in the evolution of the plume. Indeed, the lateral development of the upwind edge of the plume in Fig. S4 suggests lateral development of the fire, similar to that associated with VLS in other fires (McRae et al. 2015)."

In the discussion: "this study did not find strong evidence of the effect of VLS on fire behaviour, as indicated by small patches of VLS-prone areas near Dunalley township (Figure S4). It is therefore possible that the pyroCb attained its maximum height without VLS. However, this interpretation should be taken with caution as VLS possibly occurred but data constraints (especially the spatial resolution of the DEM and wind direction) may have precluded accurate determination of VLS".

RC: 29. The authors should cite Terrasson et al. here

AC: The article has now been cited.

RC: 30. In Terrasson et al. (2019) we did exactly that, e.g. linking fire behaviour to plume development. The authors could also cite McCarthy et al. (2018) where fire behaviour is studied against plume development for a PyroCu in Victoria (Mt Bolton fire). McCarthy, N., H. McGowan, A. Guyot, and A. Dowdy, 2018: Mobile X-Pol Radar :

[Figure]

A New Tool for Investigating Pyroconvection and Associated Wildfire Meteorology. Bull. Amer. Meteor. Soc., 99, 1177–1195, https://doi.org/10.1175/BAMS-D-16-0118.1

AC: We have included the below sentence in the weather radar paragraph of the discussion: "A feature of our study was linking plume evolution to fire severity mapping - an approach that has received limited attention. Duff et al. (2018) conducted one of the few studies, using statistical models to link the radar-detected plume volume to fire growth, and found that radar return volume (above a threshold of 10 dBZ) was a robust predictor of fire-area change. . . . [sentence removed] . . . . . . . . .. Other Australian studies linking fire behaviour to radar-detected plume development include McCarthy et al. (2018) and Terrasson et al. (2019)".

RC: 31. Fig.S3: "the orange colour represents the most intense parts of the PyroCb": it would be appropriate to use a more scientific and precise language, i.e. provide the values or range of values for the equivalent reflectivity in dBZ (colour bar), and refer to the text of the article to describe how these have to be interpreted.

AC: The colour bar has now been included in the maps, and the figure captioning updated to reflect the reflectivity values for the intense parts of the plume. We have moved the map from the supplement to to the main article in Section 3.1.3 in response to another reviewer. We have now added a sentence that describes pyroCb development in the atmosphere.

It reads: "The period of the pyroCb in Fig. 4 is defined by very high radar returns, with reflectivity values of 48-88 dBZ, representing the most intense parts of the pyroCb. This strong reflectivity is indicative of high quantities of ash, and larger-sized hydrometeors such as ice crystals in the higher elevations".

RC: Thanks and again I enjoyed reading your study. Kind regards.

AC: Thanks.
* * *
2019-354, 2019.

**(a)**

1000

0 15 30 60 90 km

**Legend**
— Ann. rainfall
■ 2013 fires
Water

**Elevation**
1 - 100
100 - 200
200 - 300
300 - 400
400 - 500
500 - 650
>650

**(b)**

(c) Rebecca White

(c) BoM, Hobart

**(c)**

N

**Legend**
▭ Fire perimeter
**Vegetation type**
■ Dry Euc. forest
■ Wet Euc. forest
■ Euc. plantation
■ Pinus plantation
■ Non-forest
■ Rainforest
■ Other

0 2 4 8 12 km

**(d)**

Dunalley

600

700

800

900

1000

Weather radar

GDA 94 MGA Zone 55

**Fig. 1.**

**Fig. 2.**

Fig. 3.

**FFDI-CH fires 2007-2016**

Daily C-Haines

Max. daily FFDI

**Sum of days per cell**  0   10K  1M  20M

**Max. fire size (ha):**  ○ 11,000  ○ 22,000  ○ 33,000  ○ 44,000  ○ 61,820

Dunalley

**Fig. 4.**

**Fig. 5.**

---

## Author Comment (AC2) · 21 Mar 2020

RC: The manuscript subject is the analysis of a pyroCb event in Tasmania, the first on record, that occurred in the Forcett-Dunalley on 2013. Its formation and evolution is related to fire danger and the C-Haines index, as well as with fire severity. I found the work solid and writing effective. The only issue of note is the role attributed to VLS (vorticity-driven lateral spread) by the authors. From the information and figures provided I really don't see enough empirical evidence for it, so I think the text could be more cautious.

AC: We appreciate your comments on VLS. See the end of this response for details on VLS.

RC: See below specific comments addressing mostly minor concerns.

RC: L11. I am not sure whether pyroCb development can be called a fire characteristic.

AC: The development of a PyroCb is contingent on the presence of a fire of sufficient intensity, therefore PyroCb development is intrinsically linked to fire behaviour, and the drivers of that fire behaviour. The generation of local weather associated with PyroCb in turn generates a feedback that further alters fire behaviour, so the linkage between the two phenomena is clear, and we believe studying PyroCb as a characteristic aspect of fire is justified.

RC: L31. 'energetically intense". Fire intensity is energy by definition. Maybe rephrase to 'high-intensity' or similar.

AC: We have rephrased the sentence to "Anthropogenic climate change is increasing the occurrence of dangerous fire weather conditions globally, leading to high-intensity wildland fires".

RC: L32. Shouldn't it be 'conducive to'?

AC: We have rephrased the sentence to ". . .climate projections suggest a pronounced increased risk of extreme fire events in Australia, with 15-70% increase in number of days conducive to extreme wildfire by 2050 in most locations. . ."

RC: L35. Check ';', it's breaking the sentence flow.

AC: The sentence has been revised, to also include suggestions by another reviewer.

It now reads: "While fire weather is most often understood as a surface phenomenon (for example, through surface temperature, wind speed and relative humidity), atmospheric processes such as instability, wind shear and mesoscale conditions can also drive extreme fire development."

RC: L35-36. Temperature is essentially expressed through fuel dryness. So remove the former or remove the later and add relative humidity.

AC: The near-surface fire danger (FFDI) is computed from temperature, RH, wind speed and fuel dryness. However, to avoid doubt and reduce the wording, we have replaced fuel dryness with relative humidity.

It now reads as follows: "While fire weather is most often understood as a surface phenomenon (for example, through surface temperature, wind speed and relative humidity), atmospheric processes such as instability, wind shear and mesoscale conditions can also drive extreme fire development".

RC: L37. This is a narrow definition of an extreme fire, as deep flaming and atmosphere coupling are often absent. Better to rephrase to make more explicit what the authors consider extreme, i.e. a particular type or range of extreme fire behaviour.

AC: We have rephrased the sentence to: "Definitions of extreme wildfires vary (e.g. Sharples et al., 2016), but associated behaviour includes rapid spread (>50 m min-1), high fireline intensity (>10,000 kW m-1), long distance spotting, erratic behaviour, and impossibility of control, often with associated development of violent pyroconvection (Tedim et al., 2018). In some cases, violent pyroconvection can manifest as pyrocumulonimbus clouds (pyroCb), the tops of which can reach the upper troposphere and lower stratosphere, ….".

RC: L46. 'Near the pyroCb' suggests it is near the surface.

AC: Agreed, it is both near the surface and in the vicinity of the pyroCb. Gust outflows could also endanger firefighters not immediately below the pyroCb. So, we prefer to keep the text as it is.

RC: L64. More accurately, the C-Haines index indicates the potential for large fire development.

AC: Agreed. However, our definition ('C-Haines index provides a measure of the potential for erratic fire behaviour') is also correct, similar to Potter (2018) below. So, we have decided to keep the text as it is. Potter, B. 2018. The Haines Index – it's time to

revise it or replace it. International Journal of Wildland Fire 27:437-440.

RC: L65. From 0 to 13?

AC: Theoretically, there is no upper bound on C-Haines, but realistic values of the input variables restrict its range. Mills & McCaw (2010) have reported the maximum as ∼13; Yeo et al. (2015) quote the upper bounds as 13.5 while Di Virgilio et al. (2019) report values such as 13.7. In this study, gridded CH for the period 2007-2016 reached 12.63.

The text now reads:" The C-Haines index provides a measure of the potential for erratic fire behaviour, based on air temperature lapse and moisture content between two lower-tropospheric levels, and typically ranges from 0-13, although values above 13 are possible (Yeo et al., 2015; Di Virgilio et al., 2019).

Yeo, C. S., J. D. Kepert, and R. Hicks. 2015. Fire danger indices: current limitations and a pathway to better indices., Bushfire & Natural Hazards CRC, Australia. Di Virgilio, G., J. P. Evans, S. A. P. Blake, M. Armstrong, A. J. Dowdy, J. Sharples, and R. McRae. 2019. Climate Change Increases the Potential for Extreme Wildfires. Geophys. Res. Lett. 46:8517-8526.

RC: L190. 'These thresholds were chosen to define an elevated fire weather day (95th percentile)' is confusing. Maybe 'to correspond' instead of 'to define'?

AC: We have revised the sentence to: "These thresholds were defined to correspond to an elevated fire weather day (95th percentile) based on weather conditions at Hobart Airport,....".

L218. The figure would benefit from the inclusion of wind direction. Did it change noticeably during the fire duration and is there information about wind direction at different heights? This can be important in relation to the assumed/inferred VLS subsequently in the text. Changes in wind direction impact the fire plume and can change spotting patterns hence modifying fire growth rate and direction.

AC: We have now included text on and a time series plot of wind direction. The time

series plot is now separated into two: one for wind variables, the other one for the other weather variables. See Fig. 1 in this supplement for details.

RC: L239 (Figure 3). So, fire severity distribution is for three periods? Wouldn't it be possible to partition those more? This would require fire growth isochrones or perhaps take advantage of the information provided by Jon Marsden-Smedley report? Also, to supplement fire severity and context, the text could indicate (possibly in the discussion) rates of spread of the wildfire for those periods, as they are available (at least in part, did not check) in said report.

AC: The fire severity progression is only for three periods as they represent the short period of erratic fire behaviour on 4 January. Two intermediate isochrones at around 14:30 and 17:30 on 4 Jan (on Jon Marsden- Smedley's report) were obtained from fire spread modelling and were not part of the official isochrones provided by Tasmania Fire Service (TFS). So, we excluded them in the previous fire severity paper as well as this one. There are other isochrones (representing the entire fire duration, 3-18 Jan) that were omitted as we were only interested in the period of erratic fire behaviour on 4 January.

We have now included info on the rate of spread during those three periods in the discussion as follows: "Mapping of the Forcett-Dunalley fire by Ndalila et al. (2018) showed that areas subjected to the highest fire intensities broadly aligned with un-dulating terrain and long unburnt dry Eucalyptus forest which under the influence of strong winds produced an ember storm that impacted the coastal township of Dunal-ley situated in the lee of the low hills. During this period (at 15:25 on 4 January), the rate of fire spread was reported to be around 50 m min-1 (or 3 km h-1), which then reduced to 1.9 km h-1 between 17:30 and 20:00, and by the time of the next fire isochrone, when fire severity and area burnt had significantly reduced (Fig. 3), the rate of spread was 1 km h-1 (Marsden-Smedley, 2014). It must be acknowledged that the role of downdrafts and mass spotting underneath the plume during the period of extreme fire behaviour is hard to infer without additional data sources on fire behaviour

(such as infra-red/multispectral linescans) and high-resolution coupled fire-atmosphere modelling (Peace et al., 2015)".

RC: L255. I know it is given in another paper, but readers would benefit from additional information about fire severity in the Methods section.

AC: Noted. We have now included additional information in the methods section as follows: "The chronosequence of fire severity was derived from a previous study (Ndalila et al., 2018) from the intersection of fire severity map with fire progression isochrones within the fireground. Fire severity was based on differential normalised burn ratio (dNBR; Key and Benson, 2006) analysis of pre- and post-fire 30 m resolution Landsat 7 satellite images. A detailed description of fire severity assessments is provided by Ndalila et al. (2018)".

RC: L256. See comment regarding L239.

AC: As aforementioned, we decided to only use isochrones that were provided by TFS. Our interpretation is therefore based on those isochrones.

RC: L291. 500 ha is not that high as a large fire size threshold. I would appreciate an enhancement of this figure where the green dots were attributed different colours corresponding to different fire size classes.

AC: We have now differentiated the fires into different fire size classes and represented them as different-sized dots, with larger dots in the graph corresponding to large fires sizes. See Fig. 2 in for details.

RC: L327. 'satellite fires'?

AC: For clarity, we have rephased it to 'new fires'.

RC: L425. And yet, Phoenix does not incorporate fire-atmosphere relationships, right?

AC: Yes. It only models fire behaviour on the surface (rate of fire spread, fire intensity, spotting etc). However, when the fire is coupled to the atmosphere, it fails to model the

feedback, which needs more advanced modelling framework such as the atmospheric model Weather & Forecasting (WRF) which is coupled to a fire spread model. Peace et al. (2015) simulated fire behaviour in Kangaroo Island, south Australia using the mentioned modelling framework.

RC: L436. The interactions with terrain were suggested, rather than shown. From what I see in the supplementary material the VLS areas were quite small and fragmented in the landscape.

AC: We agree that the VLS-prone areas seem small, and their influence on pyroCb occurrence likely negligible based on these results, but we also caution that VLS likely occurred, but the effect may have been undermined by the limitations of the input data (DEM and wind direction layer). We have now rephrased the sentence in both the results and discussion sections to read:

In section 3.1.3 (Results): "Evidence of an effect of Vorticity-driven Lateral Spread on the fire behaviour was not strong. Analysis of the precursor terrain conditions only revealed small patches of VLS-prone areas near Dunalley township (Fig. S4). However, we are not able to rule out VLS occurrence on parts of the terrain that were not resolved by the DEM, and which may have played a part in the evolution of the plume. Indeed, the lateral development of the upwind edge of the plume in Fig. S4 suggests lateral development of the fire, similar to that associated with VLS in other fires (McRae et al. 2015)."

In the discussion: "this study did not find strong evidence of the effect of VLS on fire behaviour, as indicated by small patches of VLS-prone areas near Dunalley township (Figure S4). It is therefore possible that the pyroCb attained its maximum height without VLS. However, this interpretation should be taken with caution as VLS possibly occurred but data constraints (especially the spatial resolution of the DEM and wind direction) may have precluded accurate determination of VLS".

RC: Enjoyed reading the manuscript.

AC: Thanks!

**(a)**

Fire starts

PyroCb event

**(b)**

Radar

Radar

**Fig. 1.**

**FFDI-CH fires 2007-2016**

Daily C-Haines

Max. daily FFDI

Dunalley

**Sum of days per cell** 0 10K 1M 20M

**Max. fire size (ha):** 11,000 22,000 33,000 44,000 61,820

**Fig. 2.**

---

## Author Comment (AC3) · 21 Mar 2020

RC: General comments:

The manuscript entitled "Evolution of an extreme Pyrocumulonimbus-driven wildfire event in Tasmania, Australia" provides an analysis of the extreme fire event in Tasmania of 4 January 2013. The temporal evolution of the PyroCb is assessed using weather radar data and is then analyzed together with the McArthur Forest Fire Danger Index (FFDI), the Continuous Haines Index (C-Haines) and fire severity maps. Overall, the presentation quality is good, the results are well described and discussed, and the manuscript covers a relevant topic that has been gaining increasing attention over the last years. As such I believe that a revised version of the manuscript will present a

good contribution to the fire community.

RC: My main concern is regarding section 3.2.1 analysis of large wildfires (>500 ha) in the context of FFDI and C-Haines. Reasoning only about the number of large fires in the FFDI/C-Haines domain (Figure 5) can be misleading since, for example, when comparing the number of large fires for FFDI >25 or <25, as the number of days with FFDI < 25 is much higher, the interpretation of the figure may be biased. I would suggest plotting the smaller fires in other colour and computing for each of the 4 regions (delimited by the dashed lines) in Figure 5, the fraction of fires exceeding 500 ha. Reasoning about the fraction of large fires may be more insightful for the discussion.

AC: All the fires in the plot are >500 ha in size, irrespective of which region (of the 4) in the graph they fall. We have now differentiated the fires into different fire-size classes and represented them as different-sized dots, with larger dots in the graph corresponding to large fires sizes. See the updated graph in Fig. 1 attached.

Fire size is used as a proxy for intensity, based on the assumption a large fire will contain some high intensity areas likely to spawn a pyroCb. There is evidence to suggest that pyroCb development is more likely when expansive flaming takes place (Badlan et al., 2017), so fire size will also relate to this aspect of development. Badlan, R. L., Sharples, J. J., Evans, J. P., and McRae, R. H. D.: The role of deep flaming in violent pyroconvection, in: MODSIM 2017, 22nd International Congress on Modelling and Simulation. December 2017, edited by: Syme, G., Hatton MacDonald, D., Fulton, B. and Piantadosi, J. , Modelling and Simulation Society of Australia and New Zealand, 1090-1096, 2017.

RC: Specific comments:

RC: L65: Is the range 0-13 correct? In Figure 4 the last class has a range of 12-14 and

AC: Theoretically, there is no upper bound on C-Haines, but realistic values of the input variables restrict its range. Mills & McCaw (2010) have reported the maximum

as ~13; Yeo et al., 2015 quote the upper bounds as 13.5 while Di Virgilio et al., 2019 report values such as 13.7. In this study, gridded CH for the period 2007-2016 reached 12.63.

The text now reads: "The C-Haines index provides a measure of the potential for erratic fire behaviour, based on air temperature lapse and moisture content between two lower-tropospheric levels, and typically ranges from 0-13, although values above 13 are possible (Yeo et al., 2015; Di Virgilio et al., 2019)".

Yeo, C. S., J. D. Kepert, and R. Hicks. 2015. Fire danger indices: current limitations and a pathway to better indices., Bushfire & Natural Hazards CRC, Australia. Di Virgilio, G., J. P. Evans, S. A. P. Blake, M. Armstrong, A. J. Dowdy, J. Sharples, and R. McRae. 2019. Climate Change Increases the Potential for Extreme Wildfires. Geophys. Res. Lett. 46:8517-8526.

RC: L367 refers values up to 13.7.

AC: See the above explanation on C-Haines.

RC: L165: 2.3? Also, a few sentences explaining the intuition of the VLS could be helpful.

AC: We have added the following sentence in that section: "VLS is an atypical fire spread arising from the interaction between strong winds and terrain which creates lee-slope eddies that interact with the fire to cause lateral fire propagation, an increase in fire intensity and mass spotting downwind of the lateral spread zone".

RC: Figure 3: What are the units for Max.height?

AC:It is in km. See the y-axis label that has perimeter, length and height in km.

RC: L255: What is the actual proportion for the 3 groups? It's hard to tell just from the image.

AC: The first isochrone contributed 10% of total area burnt across all vegetation types

within the entire fire perimeter, while the peak fire period accounted for 46%, with the last period on 4 January contributing 9%. We have now included the following sentence:

"The isochrone leading up to the peak in fire behaviour accounted for 10 % of total area burnt across all vegetation types within the entire fire perimeter, the peak period contributed 46%, while the last isochrone on 4 January contributed 9 % of total area burnt (Fig. 3c)".

RC: L265: The 0.5 correlation is for 1-day lag?

AC: No. Correlation has been computed without considering the lag between the two variables.

RC: L282: It may be relevant to also comment on how extreme are these values regarding all days, for example, what percentile they correspond, 95, 99?

AC: From the reanalysis data for the period 2007-2016, the Forcett-Dunalley fire represents 99th percentile of daily C-Haines and FFDI of all the days within that specific cell in Dunalley which had the highest daily FFDI. We have not computed the percentile of all days within the entire island of Tasmania because the climate is different all over the island.

The sentence now reads: "The Forcett-Dunalley fire had amongst the highest levels of elevated fire weather (gridded FFDI and C-Haines of 68 and 11.5, respectively) of all the 77 large (>500 ha) Tasmanian fires that occurred between 2007-2016 (Fig. 5). These values represent the 99th percentile of daily FFDI and C-Haines for the grid cell in Dunalley that had the highest daily FFDI during the fire".

RC: L363: Is this true for all fires or for all large fires (>500 ha)?

AC: Yes. The Forcett-Dunalley fire had amongst the highest levels of elevated fire weather of all the large (>500 ha) Tasmanian fires (shown in Figure 3) between 2007-2016 and was the only event to have produced a pyroCb.

[Figure]

Please also note the supplement to this comment:
https://www.nat-hazards-earth-syst-sci-discuss.net/nhess-2019-354/nhess-2019-354-AC3-supplement.pdf

————————————————

**FFDI-CH fires 2007-2016**

Dunalley

Daily C-Haines

Max. daily FFDI

Sum of days per cell    0    10K  1M  20M

Max. fire size (ha):    11,000    22,000    33,000    44,000    61,820

**Fig. 1.**

---

## Author Comment (AC4) · 21 Mar 2020

RC: This paper provides an analysis of a significant pyroCb event in Tasmania (January 2013). The evolution of the plume is assessed using weather radar. Analysis of meteorology is largely based on the lower troposphere using the C-Haines index and FFDI. Fire severity maps are also included. The manuscript is generally well-organized and easy to read. Results are relevant to several research communities. The pyroCb phenomenon has been gaining significant attention in recent years. However, a few aspects of the paper require some clarification prior to publication.

RC: Please see my comments below.

RC: Abstract line 18: "highly unstable atmosphere" do you mean "highly unstable lower

troposphere" since you are referencing C-Haines?

AC: We meant highly unstable lower atmosphere i.e both the lower troposphere (C-Haines) and the low-mid- tropospheric instability (850-500 hPa in Fig. S2). We have now rephrased the sentence to: "The pyroCb was associated with a highly unstable lower atmosphere (C-Haines 10-11) . . ."

RC: Lines 65-70: lapse rate in lower-troposphere, mid-levels, or both?

AC: The C-Haines is calculated from two lower-tropospheric levels. We have now rephrased the sentence as follows: "The C-Haines index provides a measure of the potential for erratic fire behaviour, based on air temperature lapse and moisture content between two lower-tropospheric levels, and typically ranges from 0-13, although values above 13 are possible (Yeo et al., 2015; Di Virgilio et al., 2019)".

RC: Line 139: "10 hPa". . ..what are these high altitude data used for? C-Haines is only based on the lower troposphere?

AC: We were just indicating up to which atmospheric height the BARRA data were available, although we calculated up to the 150 hPa level (see the BARRA soundings in Fig. S2). We also, in Fig. S3, determined air temperature lapse rate at the 850-500 hPa, in addition to C-Haines (for 850-700 hPa height).

We have rephrased the sentence to: "We extracted hourly air temperature and dew-point temperature at different air pressure levels (1000 hPa at the surface to 150 hPa in the lower stratosphere), as well as pre-calculated hourly McArthur FFDI for the period of the fire and the period of BARRA data available at the time of the study (January 2007-October 2016)".

RC: Line 181: is there a reference for "typically happens at or near the start of fires"?

AC: We have included an example of a fire whose major run was at the start of the fire. The sentence now reads:

"A total of 77 fires, of varying ignition sources, were identified in the Tasmania Fire Service fire history database as being >500 ha in size, and having a known ignition date. Of these, 18 did not have recorded end dates, and these were operationally specified as being four weeks later, an arbitrary cut-off to capture the most likely major growth (or 'run') of the fire that typically happens at or near the start of fires (e.g. the 2009 Victorian fires; Cruz et al., 2012). The assumption is that these runs result in peak fire intensities that most likely drive strong convections".

RC: Line 195: please replace "was always higher than" with a specific range of values.

AC: The sentence now reads: "It is worth noting that maximum daily FFDI calculated at the Hobart Airport station was always higher than daily gridded FFDI extracted from the BARRA product (compare maximum FFDI of 92 recorded at the station with FFDI of 68.7 from the BARRA model on 4 January)".

RC: Section 3.1: In addition to surface fire weather, pyroCb are driven by the full 3-D structure of the troposphere. This discussion can be improved. The supplement provides thermodynamic profiles. Can you quantify the LCL and highlight the pressure levels that are used to calculate C-Haines? Was there any CAPE in these profiles? Please also provide a couple sentences describing the synoptic weather pattern that set the stage for this event.

AC: We agree with your observations that both surface fire weather and 3D troposphere influence pyroCb development. We have captured the role of these fire weather aspects in the introduction and discussion sections. In the results section, the atmospheric aspect has been described in the section on pyroCb development (Section 3.1.3), and the section on spatio-temporal variation of C-Haines. We have now included in Section 3.1.2 information about synoptic weather as per your suggestions, to further expound on the tropospheric processes associated with the event.

1. Regarding quantifying the Lifted Condensation Level, the values are now in the aerological profiles in Fig. 1 attached to this response.

2. The pressure levels used to calculate C-Haines are 850 and 700 hPa, and are shown in Equations 1-3. Full description of the variables are provided immediately below the equations.

3. CAPE is now available in the profiles. The value was zero, suggesting lower potential for thunderstorm development, with the energy to lift the air parcel and produce the pyroCb coming from the fire rather than the unmodified atmosphere. That information is provided in the supporting text in the supplement (Section S1.2) as follows:

"The Convective Available Potential Energy (CAPE) as well as the maximum unstable CAPE (MUCAPE) during this period were zero, suggesting a lower potential for thunderstorm development. It is likely that much of the energy that lifted the air parcel and produced the pyroCb came from the fire itself rather than the unmodified atmosphere".

RC: Section 3.1.2: Please show a basic radar image/map at the peak of the pyroCb event for reader orientation.

AC: We have moved the 3D radar image from the supplement to the main article and provided a short interpretation of the pyroCb as shown in the radar image, as follows:

"The period of the pyroCb in Fig. 4 is defined by very high radar returns, with reflectivity values of 48-88 dBZ, representing the most intense parts of the pyroCb. This strong reflectivity is indicative of high quantities of ash, and larger-sized hydrometeors such as ice crystals in the higher elevations"....

RC: Line 233: please briefly describe the significance of the 850-500 lapse rate in the main text.

AC: We have added the following text: "The 850-500 hPa lapse rate gives an indication of the (in)stability of the lower half of the troposphere (1.3-5.5 km above sea level), with lapse rates of >7.5 oC km-1 considered as very unstable lower atmosphere (Peterson et al., 2014)".

RC: Line 265: "good correlation of 0.5". . ..what is good?

AC: We have replaced "good correlation" with "moderate correlation". The sentence now reads: "...for the entire month of January 2013, there was a statistically significant moderate correlation (r = 0.5, p <0.05) between these fire weather indices, although FFDI lagged C-Haines by around a day".

RC: Line 269: "wind change at around 00:00"...is this a cold front? Some description is likely needed?

AC: Yes, the wind change was as a result of a pre-frontal trough that passed before 00:00 on 5 January, well after the pyroCb had already established and dissipated. We have also included information on the synoptic weather in the section (3.1.2) preceding this section to provide context.

The sentence now reads: "In the Tasman Peninsula, especially, C-Haines was mostly in the range of 10-12 for both days (peaking at 12-14) but moderated to 4-6 on 5 January after a southwest wind change at around 00:00, as a result of a pre-frontal trough crossing south-eastern Tasmania".

RC: Section 4: Did you examine additional radar products for this case, such as velocity data? You might be able to get more information on the dynamics of the updraft and even particle characteristics. Radar data are one of the more underutilized tools in pyroCb research.

AC: We greatly appreciate the reviewer's comments about making further use of the radar data. However, we feel that including all of the work suggested by the reviewer would go well beyond the intended scope of the manuscript, which was to provide a qualitative description of the evolution of the Forcett-Dunalley fire and pyroCb, and their relation to coincident surface weather and lower atmospheric conditions.

Indeed, we feel that it would be more appropriate to pursue the work suggested by the reviewer in a separate and more detailed quantitative analysis, drawing upon the work of Terrasson et al. (2019) and McCarthy et al. (2019). We have amended the

discussion to specifically note this as a natural extension of the qualitative analysis presented in the manuscript.

The section in the discussion now reads: "... radars remain a reliable data source that can provide near-real time monitoring of strong pyroconvection, as evidenced by previous pyroCb studies in Australia and globally (Rosenfeld et al., 2007; Fromm et al., 2012; Lareau and Clements, 2016; Dowdy et al., 2017; Peace et al., 2017; Lareau et al., 2018; Terrasson et al., 2019). This study did not analyse radial velocity from the Doppler radar; therefore future research on the Forcett-Dunalley fire and other fires should consider using that information to provide a more quantitative analysis of the thunderstorm, drawing upon previous work in Australia (McCarthy et al., 2019; Terrasson et al., 2019). A feature of our study was linking plume evolution to fire severity mapping - an approach that has received limited attention".

RC: Lines 335-345: The soundings in the supplement do show the inverted-V structure typical of high-based convection, with some mid-level moisture in the profile at 1500. You can calculate total precipitable water to confirm. I believe that Peterson et al. provide more info on this.

AC: The information on total precipitable water (and other metrics) is now provided in the profiles and in the discussion of the main article. The sentence in the discussion now reads:

"... a number of Australian pyroCb events exhibit a distinct lack of midlevel moisture in their associated atmospheric profiles (e.g. the 2003 Canberra fire (Fromm et al., 2006) and the 2013 Wambelong fire (Wagga Wagga sounding for 13 January 2013 on http://weather.uwyo.edu/upperair/sounding.html). However, in this study, the mid- to upper-level moisture was higher during the time preceding pyroCb formation on 4 January (Fig. S2a), with a total precipitable water of 23 mm indicating a moist lower atmosphere (Webb and Fox-Hughes, 2015). Terrasson et al. (2019) report on the effect of a change in moisture between the low- and upper-levels (brought by a cold front) on

the development of a pyroCb and enhancement of fire behaviour in the Sir Ivan fire in eastern Australia. Further research is required to properly understand the potential influence of mid-tropospheric moisture in driving pyroCb development in Australia".

RC: Supplement: Please annotate the time of the pyroCb on the relevant figures.

AC: The timestamps with the pyroCb are now shown in the radar images (Fig. 2 in this supplement below), and the time of the pyroCb is mentioned in the figure captioning (now moved to main article) as follows:

"Areas prone to vorticity-driven lateral spread (VLS, in red) overlaid with a 3D rendering of the vertical cross-section of plumes from the radar reflectivity for specific times during peak fire behaviour. Higher reflectivity values (dBZ >48) in all maps represent the most intense parts of the plume (and the pyroCb which occurred from around 15:24 to 16:30 LT). The asterisk in the 15:24 map represents the likely initiation period of the pyroCb. Dunalley township is represented by a white star. A malfunction in radar for the 15:00 and 15:36 timestamps resulted in plume information to be only available at the lowest elevation angle of the radar scan, and are therefore not shown".

RC: Supplement: Define/explain "BARRA pseudo-soundings"

AC: Pseudo-soundings are atmospheric soundings obtained from the air and dewpoint temperatures produced by the BARRA model. They are not actual weather balloon (radiosonde) observations. We have now clarified in the supplement that the pseudo-soundings are vertical profiles sampled from the reanalysis data.

RC: It would be nice to include a surface and upper level synoptic weather map that coincides with the soundings in Figure S1. This will provide some context on the large-scale meteorology driving this event.

AC: We have now provided the mean surface level pressure charts in the supplement (Fig. 3 below), and information on the synoptic weather on 3-4 January in Section 3.1.2 of the main article, which reads as follows:

[Figure]

On 3 January, a combination of: (1) a high-pressure system to the northeast of Tasmania and (2) a cold front and pre-frontal trough approaching from the west directed a freshening dry and hot northerly airstream over the island, favourable conditions for elevated fire danger (Bureau of Meteorology, 2013). By 08:00 LT on 4 January, the high-pressure system had moved only slowly eastward while the trough had progressed closer to western Tasmania (Fig. S1). This period coincided with moderate north-westerly winds in most locations in the state, except for the southeast (general area surrounding Dunalley) which recorded stronger winds and elevated fire danger. Fire danger steadily increased towards midday (the start of the first fire progression isochrone in Fig. 3c) and by 15:00, the leading edge of the trough was much closer to the west of Tasmania. An increase in pressure gradient brought about gusty conditions and catastrophic fire danger in some locations in southeast Tasmania, causing erratic fire behaviour on the Forcett-Dunalley fire. During that time, the trough crossed western Tasmania. By 17:00, when the pyroCb had likely dissipated (Fig. 3a), the pre-frontal trough was crossing Tasmania and fire danger subsequently reduced due to decreasing temperatures and winds. The trough continued to move eastwards and crossed southeast Tasmania after 23:00, leading to a west to southwest wind change by 00:00 on 5 January. The front passed over the state early morning of 5 January and caused lightning and limited showers across Tasmania. Detailed analysis of the synoptic weather patterns driving this event are provided in Bureau of Meteorology (2013).

RC: Are the times listed local time or UTC?

AC: They are in LT. This information has now been updated in the relevant sections in the supplement and main article.
* * *
[Figure]

[Figure]

**Fig. 1.**

[Figure]

Fig. 2.

[Figure]

Fig. 3.

---

## Author Response (AR1)

**Response to referee reports**

Referee 1- Paulo Fernandes

The manuscript subject is the analysis of a pyroCb event in Tasmania, the first on record, that occurred in the Forcett-Dunalley on 2013. Its formation and evolution is related to fire danger and the C-Haines index, as well as with fire severity. I found the work solid and writing effective. The only issue of note is the role attributed to VLS (vorticity-driven lateral spread) by the authors. From the information and figures provided I really don't see enough empirical evidence for it, so I think the text could be more cautious.
See the end of your review for details on VLS.

See below specific comments addressing mostly minor concerns.
L11. I am not sure whether pyroCb development can be called a fire characteristic.
The development of a PyroCb is contingent on the presence of a fire of sufficient intensity, therefore PyroCb development is intrinsically linked to fire behaviour, and the drivers of that fire behaviour. The generation of local weather associated with PyroCb in turn generates a feedback that further alters fire behaviour, so the linkage between the two phenomena is clear, and we believe studying PyroCb as a characteristic aspect of fire is justified.

L31. 'energetically intense". Fire intensity is energy by definition. Maybe rephrase to 'high-intensity' or similar.
We have rephrased the sentence to "Anthropogenic climate change is increasing the occurrence of dangerous fire weather conditions globally, leading to high-intensity wildland fires"

L32. Shouldn't it be 'conducive to'?
We have rephrased the sentence to "…climate projections suggest a pronounced increased risk of extreme fire events in Australia, with 15-70% increase in number of days conducive to extreme wildfire by 2050 in most locations…"

L35. Check ';', it's breaking the sentence flow. The sentence has been revised, to also include suggestions by another reviewer.
It now reads: "While fire weather is most often understood as a surface phenomenon (for example, through surface temperature, wind speed and relative humidity), atmospheric processes such as instability, wind shear and mesoscale conditions can also drive extreme fire development."

L35-36. Temperature is essentially expressed through fuel dryness. So remove the former or remove the later and add relative humidity.
The near-surface fire danger (FFDI) is computed from temperature, RH, wind speed and fuel dryness. However, to avoid doubt and reduce the wording, we have replaced fuel dryness with relative humidity.

It now reads as follows: "While fire weather is most often understood as a surface phenomenon (for example, through surface temperature, wind speed and relative humidity), atmospheric processes such as instability, wind shear and mesoscale conditions can also drive extreme fire development".

L37. This is a narrow definition of an extreme fire, as deep flaming and atmosphere coupling are often absent. Better to rephrase to make more explicit what the authors consider extreme, i.e. a particular type or range of extreme fire behaviour.

We have rephrased the sentence to: "Definitions of extreme wildfires vary (e.g. Sharples et al., 2016), but associated behaviour includes rapid spread (>50 m min-1), high fireline intensity (>10,000 kW m-1), long distance spotting, erratic behaviour, and impossibility of control, often with associated development of violent pyroconvection (Tedim et al., 2018). In some cases, violent pyroconvection can manifest as pyrocumulonimbus clouds (pyroCb), the tops of which can reach the upper troposphere and lower stratosphere, ….".

L46. 'Near the pyroCb' suggests it is near the surface. Agreed, it is both near the surface and in the vicinity of the pyroCb. Gust outflows could also endanger firefighters not immediately below the pyroCb. So, we prefer to keep the text as it is.

L64. More accurately, the C-Haines index indicates the potential for large fire development.

Agreed. However, our definition ('C-Haines index provides a measure of the potential for erratic fire behaviour') is also correct, similar to Potter (2018) below. So, we have decided to keep the text as it is.

Potter, B. 2018. The Haines Index – it's time to revise it or replace it. *International Journal of Wildland Fire* **27**:437-440.

L65. From 0 to 13?

Theoretically, there is no upper bound on C-Haines, but realistic values of the input variables restrict its range. Mills & McCaw (2010) have reported the maximum as ~13; Yeo et al., 2015 quote the upper bounds as 13.5 while Di Virgilio et al., 2019 report values such as 13.7. In this study, gridded CH for the period 2007-2016 reached 12.63.

The text now reads:" The C-Haines index provides a measure of the potential for erratic fire behaviour, based on air temperature lapse and moisture content between two lower-tropospheric levels, and typically ranges from 0-13, although values above 13 are possible (Yeo et al., 2015; Di Virgilio et al., 2019).

Yeo, C. S., J. D. Kepert, and R. Hicks. 2015. Fire danger indices: current limitations and a pathway to better indices., Bushfire & Natural Hazards CRC, Australia.
Di Virgilio, G., J. P. Evans, S. A. P. Blake, M. Armstrong, A. J. Dowdy, J. Sharples, and R. McRae. 2019. Climate Change Increases the Potential for Extreme Wildfires. *Geophys. Res. Lett.* **46**:8517-8526.

L190. 'These thresholds were chosen to define an elevated fire weather day (95th percentile)' is confusing. Maybe 'to correspond' instead of 'to define'?

We have revised the sentence to:

"These thresholds were defined to correspond to an elevated fire weather day (95th percentile) based on weather conditions at Hobart Airport,….".

L218. The figure would benefit from the inclusion of wind direction. Did it change noticeably during the fire duration and is there information about wind direction at different heights? This can be important in relation to the assumed/inferred VLS subsequently in the text. Changes in wind direction impact the fire plume and can change spotting patterns hence modifying fire growth rate and direction.

Wind direction has now been included, and the time series plot separated into two: one for wind variables, the other one for the other weather variables.

 So, fire severity distribution is for three periods? Wouldn't it be possible to partition those more? This would require fire growth isochrones or perhaps take advantage of the information provided by Jon Marsden-Smedley report? Also, to supplement fire severity and context, the text could indicate (possibly in the discussion) rates of spread of the wildfire for those periods, as they are available (at least in part, did not check) in said report.

The fire severity progression is only for three periods as they represent the short period of erratic fire behaviour on 4 January. Two intermediate isochrones at around 14:30 and 17:30 on 4 Jan (on Jon Marsden- Smedley's report) were obtained from fire spread modelling and were not part of the official isochrones provided by Tasmania Fire Service (TFS). So, we excluded them in the previous fire severity paper as well as this one. There are other isochrones (representing the entire fire duration, 3-18 Jan) that were omitted as we were only interested in the period of erratic fire behaviour on 4 January.

**We have now included info on the rate of spread during those three periods in the discussion as follows:**
"Mapping of the Forcett-Dunalley fire by Ndalila et al. (2018) showed that areas subjected to the highest fire intensities broadly aligned with undulating terrain and long unburnt dry *Eucalyptus* forest which under the influence of strong winds produced an ember storm that impacted the coastal township of Dunalley situated in the lee of the low hills. During this period (at 15:25 on 4 January), the rate of fire spread was reported to be around 50 m min$^{-1}$ (or 3 km h$^{-1}$), which then reduced to 1.9 km h$^{-1}$ between 17:30 and 20:00, and by the time of the next fire isochrone, when fire severity and area burnt had significantly reduced (Fig. 3), the rate of spread was 1 km h$^{-1}$ (Marsden-Smedley, 2014). It must be acknowledged that the role of downdrafts and mass spotting underneath the plume during the period of extreme fire behaviour is hard to infer without additional data sources on fire behaviour (such as infra-red/multispectral linescans) and high-resolution coupled fire-atmosphere modelling (Peace et al.,

2015)".

 I know it is given in another paper, but readers would benefit from additional information about fire severity in the Methods section.
Noted. We have now included additional information in the methods section as follows:
"The chronosequence of fire severity was derived from a previous study (Ndalila et al., 2018) from the intersection of fire severity map with fire progression isochrones within the fireground. Fire severity was based on differential normalised burn ratio (dNBR; Key and Benson, 2006) analysis of pre- and post-fire 30 m resolution Landsat 7 satellite images. A detailed description of fire severity assessments is provided by Ndalila et al. (2018)".

 As aforementioned, we decided to only use isochrones that were provided by TFS. Our interpretation is therefore based on those isochrones.

ha is not that high as a large fire size threshold. I would appreciate an enhancement of this figure where the green dots were attributed different colours corresponding to different fire size classes.

We have now differentiated the fires into different fire size classes and represented them as different-sized dots, with larger dots in the graph
corresponding to large fires sizes

L327. 'satellite fires'? For clarity, we have rephased it to 'new fires'.

L425. And yet, Phoenix does not incorporate fire-atmosphere relationships, right?
Yes. It only models fire behaviour on the surface (rate of fire spread, fire intensity, spotting etc). However, when the fire is coupled to the
atmosphere, it fails to model the feedback, which needs more advanced modelling framework such as the atmospheric model Weather & Forecasting
(WRF) which is coupled to a fire spread model. Peace et al. (2015) simulated fire behaviour in Kangaroo Island, south Australia using the mentioned
modelling framework.

L436. The interactions with terrain were suggested, rather than shown. From what I see in the supplementary material the VLS areas were quite
small and fragmented in the landscape.
We agree that the VLS-prone areas seem small, and their influence on pyroCb occurrence likely negligible based on these results, but we also
caution that VLS likely occurred, but the effect may have been undermined by the limitations of the input data (DEM and wind direction layer).

We have now rephrased the sentence in both the results and discussion sections to read:
In section 3.1.3 (Results): "Evidence of an effect of Vorticity-driven Lateral Spread on the fire behaviour was not strong. Analysis of the precursor
terrain conditions only revealed small patches of VLS-prone areas near Dunalley township (Fig. S4). However, we are not able to rule out VLS
occurrence on parts of the terrain that were not resolved by the DEM, and which may have played a part in the evolution of the plume. Indeed, the
lateral development of the upwind edge of the plume in Fig. S4 suggests lateral development of the fire, similar to that associated with VLS in other
fires (McRae et al. 2015)."

In the discussion: "this study did not find strong evidence of the effect of VLS on fire behaviour, as indicated by small patches of VLS-prone areas
near Dunalley township (Figure S4). It is therefore possible that the pyroCb attained its maximum height without VLS. However, this interpretation
should be taken with caution as VLS possibly occurred but data constraints (especially the spatial resolution of the DEM and wind direction) may
have precluded accurate determination of VLS".

Enjoyed reading the manuscript. Thanks!

**Referee 2- Anonymous**

General comments

The manuscript entitled "Evolution of an extreme Pyrocumulonimbus-driven wildfire event in Tasmania, Australia" provides an analysis of the extreme fire event in Tasmania of 4 January 2013. The temporal evolution of the PyroCb is assessed using weather radar data and is then analyzed together with the McArthur Forest Fire Danger Index (FFDI), the Continuous Haines Index (C-Haines) and fire severity maps. Overall, the presentation quality is good, the results are well described and discussed, and the manuscript covers a relevant topic that has been gaining increasing attention over the last years. As such I believe that a revised version of the manuscript will present a good contribution to the fire community.

My main concern is regarding section 3.2.1 analysis of large wildfires (>500 ha) in the context of FFDI and C-Haines. Reasoning only about the number of large fires in the FFDI/C-Haines domain (Figure 5) can be misleading since, for example, when comparing the number of large fires for FFDI >25 or <25, as the number of days with FFDI < 25 is much higher, the interpretation of the figure may be biased. I would suggest plotting the smaller fires in other colour and computing for each of the 4 regions (delimited by the dashed lines) in Figure 5, the fraction of fires exceeding 500 ha. Reasoning about the fraction of large fires may be more insightful for the discussion.

All the fires in the plot are >500 ha in size, irrespective of which region (of the 4) in the graph they fall. We have now differentiated the fires into different fire-size classes and represented them as different-sized dots, with larger dots in the graph corresponding to large fires sizes.

Fire size is used as a proxy for intensity, based on the assumption a large fire will contain some high intensity areas likely to spawn a pyroCb. There is evidence to suggest that pyroCb development is more likely when expansive flaming takes place (Badlan et al., 2017), so fire size will also relate to this aspect of development.

Specific comments:

L65: Is the range 0-13 correct? In Figure 4 the last class has a range of 12-14 and

Theoretically, there is no upper bound on C-Haines, but realistic values of the input variables restrict its range. Mills & McCaw (2010) have reported the maximum as ~13; Yeo *et al.*, 2015 quote the upper bounds as 13.5 while Di Virgilio *et al.,* 2019 report values such as 13.7. In this study, gridded CH for the period 2007-2016 reached 12.63.

The text now reads: "The C-Haines index provides a measure of the potential for erratic fire behaviour, based on air temperature lapse and moisture content between two lower-tropospheric levels, and typically ranges from 0-13, although values above 13 are possible (Yeo et al., 2015; Di Virgilio et al., 2019)".

Yeo, C. S., J. D. Kepert, and R. Hicks. 2015. Fire danger indices: current limitations and a pathway to better indices., Bushfire & Natural Hazards CRC, Australia.

Di Virgilio, G., J. P. Evans, S. A. P. Blake, M. Armstrong, A. J. Dowdy, J. Sharples, and R. McRae. 2019. Climate Change Increases the Potential for

Extreme Wildfires. *Geophys. Res. Lett.* **46**:8517-8526.

L367 refers values up to 13.7. See the above explanation.

L165: 2.3? Also, a few sentences explaining the intuition of the VLS could be helpful.

We have added the following sentence in that section:

"VLS is an atypical fire spread arising from the interaction between strong winds and terrain which creates lee-slope eddies that interact with the fire to cause lateral fire propagation, an increase in fire intensity and mass spotting downwind of the lateral spread zone".

Figure 3: What are the units for Max.height? It is in km. See the y-axis label that has perimeter, length and height in km.

L255: What is the actual proportion for the 3 groups? It's hard to tell just from the image.

The first isochrone contributed 10% of total area burnt across all vegetation types within the entire fire perimeter, while the peak fire period accounted for 46%, with the last period on 4 January contributing 9%. We have now included the following sentence:

"The isochrone leading up to the peak in fire behaviour accounted for 10 % of total area burnt across all vegetation types within the entire fire perimeter, the peak period contributed 46%, while the last isochrone on 4 January contributed 9 % of total area burnt (Fig. 3c)".

L265: The 0.5 correlation is for 1-day lag? No. Correlation has been computed without considering the lag between the two variables.

L282: It may be relevant to also comment on how extreme are these values regarding all days, for example, what percentile they correspond, 95, 99?

From the reanalysis data for the period 2007-2016, the Forcett-Dunalley fire represents 99th percentile of daily C-Haines and FFDI of all the days within that specific cell in Dunalley which had the highest daily FFDI. We have not computed the percentile of all days within the entire island of Tasmania because the climate is different all over the island.

The sentence now reads:

"The Forcett-Dunalley fire had amongst the highest levels of elevated fire weather (gridded FFDI and C-Haines of 68 and 11.5, respectively) of all the 77 large (>500 ha) Tasmanian fires that occurred between 2007-2016 (Fig. 5). These values represent the 99th percentile of daily FFDI and C-Haines for the grid cell in Dunalley that had the highest daily FFDI during the fire".

L363: Is this true for all fires or for all large fires (>500 ha)?

Yes. The Forcett-Dunalley fire had amongst the highest levels of elevated fire weather of all the large (>500 ha) Tasmanian fires (shown in Figure 3) between 2007-2016 and was the only event to have produced a pyroCb.

**Short comment- Adrien Guyot**

This article by Ndalila et al. presents the concomitant analysis of a wildfire plume, which produced a pyroCb, and surface fire behaviour. I quite enjoyed reading the paper, and I am quite familiar with the area where the fire took place. My research interests include the use of weather radar to monitor fire plumes, therefore I felt it useful to provide some comments that might help improve the paper.Overall, I feel that the radar data, possibly making half of the paper content (the other half being fire behaviour observations) are well under-utilised by the authors. For instance, I am surprised that the authors did not show and discuss radar observations of the Doppler velocity? We have shown in Terrasson et al. (2019) that these can provide great insights into the dynamics of the PyroCb, including convection, shear, vortices etc. Its derivative, i.e. the spectrum width as we showed in Terrasson et al. (2019) and McCarthy et al. (2017; 2018) can also give great insights.

Here, the radar data is only presented as a time-series of integrated variables in one figure of the article, which gives limited information. The Figure S3 in the supplementary material shows only cross sections of the Reflectivity, without colour scale, which is really limited. Recent work by McCarthy et al. (2019) and number of papers from Lareau and Clements show that radar observations can be utilised to draw multiple quantitative and qualitative information: this richness must be utilised.

We greatly appreciate the reviewer's comments about making further use of the radar data. However, we feel that including all of the work suggested by the reviewer would go well beyond the intended scope of the manuscript, which was to provide a qualitative description of the evolution of the Forcett-Dunalley fire and pyroCb, and their relation to coincident surface weather and lower atmospheric conditions. Indeed, we feel that it would be more appropriate to pursue the work suggested by the reviewer in a separate and more detailed quantitative analysis, drawing upon the work of Terrasson et al. (2019) and McCarthy et al. (2019). We have amended the discussion to specifically note this as a natural extension of the qualitative analysis presented in the manuscript.

Additionally, an aspect of the Sir Ivan fire in Terrasson et al. (2019) was that the pyroCb was a supercell, which is a more organised type of thunderstorm, with more to investigate in the radar data. There's no evidence of that in Dunalley, but this might make a similar analysis more interesting as a complement to that study.

The section in the discussion now reads: "… radars remain a reliable data source that can provide near-real time monitoring of strong pyroconvection, as evidenced by previous pyroCb studies in Australia and globally (Rosenfeld et al., 2007; Fromm et al., 2012; Lareau and Clements, 2016; Dowdy et al., 2017; Peace et al., 2017; Lareau et al., 2018; Terrasson et al., 2019). This study did not analyse radial velocity from the Doppler radar; therefore future research on the Forcett-Dunalley fire and other fires should consider using that information to provide a more quantitative analysis of the thunderstorm, drawing upon previous work in Australia (McCarthy et al., 2019; Terrasson et al., 2019). A feature of our study was linking plume evolution to fire severity mapping - an approach that has received limited attention".

Specific comments below:

1. Title: " . . .Pyrocumulonimbus-driven . . . ." We know of the feedback loop between surface fire behaviour and pyroCb, but it reads as if the fire was influenced by the PyroCb and not the other way around. What do you mean by "Extreme PyroCb", how does it differ from a standard
PyroCb? The use of more scientific rather than emotive language should be preferred.
Noted. Actually, extreme in this case describes the wildfire event, not the pyroCb. Regarding the statement that the title supposes that the pyroCb caused the extreme fire, that is also true, although we meant, as you correctly put, that the extreme fire caused the pyroCb.

A new title now reads: "Evolution of a pyrocumulonimbus event associated with an extreme wildfire in Tasmania, Australia".
2. L34 and other occurrences in the article. Shouldn't references listed in brackets be in chronological order?
Noted. All the references are now in chronological order.

3. L36: this seems somehow restrictive, as not only atmospheric instability but wind shear and mesoscale conditions can drive fire behaviour.
We have rephrased the sentence to read: "While fire weather is most often understood as a surface phenomenon (for example, through surface temperature, wind speed and relative humidity), atmospheric processes such as instability, wind shear and mesoscale conditions can also drive extreme fire development".

4. L44; Chronological order
Noted. The references are now in chronological order.

5. Melnikov et al. (2008) does not present observations of PyroCb so the reference should not be cited here. Authors should also cite Terrasson et al. (2009) in which we report detailed radar observations of a PyroCb in NSW.

Terrasson, A., McCarthy, N., Dowdy, A., Richter, H., McGowan, H., & Guyot, A. ( 2019). Weather radar insights into the turbulent dynamics of a wildfire-triggered supercell thunderstorm. Journal of Geophysical Research: Atmospheres, 124, 8645– 8658.https://doi.org/10.1029/2018JD029986
We have replaced Melnikov et al. (2008) reference with Terrasson et al. (2009).

6. L57 and 58: This sentence would probably benefit from the use of a more scientific description of the radar capabilities, e.g. . . . ."they are "tuned"
to identify" is actually incorrect. The frequency of the radar will determine which particles the radar is likely to receive a backscatter from. Post-processing analysis can be improved possibly, but it won't modify the frequency of the radar, which is a fixed hardware choice. More details on the scatterers as observed by the radar and reference to pyrometeors as described by McCarthy et al. (2019) would be appropriate here.

McCarthy, N. F., Guyot, A., Protat, A., Dowdy, A., & McGowan, H. ( 2019). Tracking Pyrometeors with Meteorological Radar Using Unsupervised Machine
Learning. Geophysical Research Letters, 46. https://doi.org/10.1029/2019GL084305

We respectfully disagree that the statement is incorrect. We however agree with you that the frequency of radar determines what kind of particles it receives backscatter from. In line 104 of our original submission, we noted that weather radars do not detect smoke components smaller than 100 μm. Therefore, we believe that the above statement is correct.

For clarity, we have replaced 'tuned to' with 'sensitive' as follows:
"Weather radars ….. do not accurately detect the exact extent of entrained gaseous and fine particulate emissions because they are sensitive to larger particles such as rain and ice crystals, and can therefore fortuitously detect pyrometeors such as ash, scorched debris and embers (McCarthy et al., 2019).

We checked the reference you provided, whose authors agree with us by suggesting that there is "increasing evidence that weather radar does not have the sensitivity to detect aerosol-sized smoke targets. Instead it detects ash and larger debris, characterized as generally horizontal, plate-like targets with high (low) ZDR ($\rho$hv)".

Additionally, the below reference states: "The consensus with such polarimetric radar results is now that the scatterers within fire plumes are
dominantly ash". The article goes on to list all scatterers such as "coarse mode particle emissions, ash, firebrands, and extinguished and scorched debris."
McCarthy, N., Guyot, A., Dowdy, A., and McGowan, H.: Wildfire and Weather Radar: A Review, J. Geophys. Res. Atmos., 124, 266-286, doi:10.1029/2018jd029285, 2019.

7. L69: the authors might want to discuss here the V-shape profile as described by Peterson et al.?
We have now added the following section:
"The role of tropospheric temperature and moisture in pyroCb dynamics is exemplified in the inverted-V thermodynamic profile (Peterson et al., 2017). The profile shows a dry and warm near-surface environment in which temperature decreases dry adiabatically with altitude to the top of the mixed layer (~ 3 km) where relative humidity is higher. Altitudes immediately above the mixed layer are usually drier, and this dry air can mix to
the surface in strong convective downdrafts, increasing surface fire behaviour (McRae et al., 2015). Further, higher mid-troposphere moisture can also interact with weaker wind shear and high temperature lapse rate to produce strong updrafts (Peterson et al., 2017)".

8. L71: it would be useful to give here the frequency of the radar (C-band) and state that this is a Doppler radar. The reader might also want to know if this is an operational radar (thus with a given scan strategy) and by whom it is operated (BoM).
We have now included the word 'Doppler' in the sentence. The other suggested information has been provided in the description of the weather radar in section 2.2.1 (in the Methods section).

9. L83: space between "600" and "m"
It has now been corrected.

10. Can you give here the time at which the pyroCb started to form?
It is difficult to know the exact time, but likely between 15:24 and 15:30. A malfunction in radar between 15:30 and 15:36 resulted in plume information to be only available at the lowest elevation angle of the radar scan, and was therefore not useful.
The sentence now reads: "On 4 January, southeast Tasmania recorded dangerous fire weather conditions, resulting in a large uncontrollable fire
that led to the development of a pyrocumulonimbus (Fig. 1Sb) from around 15:24 and caused the near-complete destruction of Dunalley township (Fig. 1d)".

11. Figure 1(d), the location of the weather radar is hard to see . . . could you possibly use another colour or/and larger font?

The colour of the symbol has been changed and the size increased.

12. L108: This is likely to be case but how did McRae validate this? What do you mean by "violent convection"? i.e. did you compute vertical velocities?

We did not compute vertical velocities, but the occurrence of the pyroCb (that rapidly rose from 9 km at 15:24 to 15 km at 15:48) is evidence of violent convection. The term 'violent convection' is used in the NWCG Glossary of Wildland Fire (dating back to 1949, most recent version is 330 2018) and has been adopted in the 2012 AFAC Bushfire Glossary. It entails a combination of strong (i.e. enough uplift to carry large particles that show up in radar data), highly turbulent and deep convection.

NWCG Glossary: https://www.nwcg.gov/glossary/a-z
AFAC Glossary: https://www.afac.com.au/docs/default-source/doctrine/bushfire-terminology.pdf

For clarity, we have, in the introduction defined 'violent convection as 'strong, highly turbulent and deep convection', and maintained the term 'violent convection' thereafter in the manuscript.

13. L111: there is no consistency with the spelling of "Mt" and "Mt." in the paper. I think it should be "Mt"?

Noted. They have been revised to "Mt" throughout the manuscript.

14. L119: Why is that threshold of 11 dBZ being used? The paper would benefit here from more explanation on the processing of radar data (clutter removal, attenuation correction if any etc.)

We used the 11 dBZ threshold as it clearly demarcated the boundary of the plume better than lower values. We did not do detailed pre-processing 345 such as what you have suggested. There is a degree of pre-processing of radar images prior to operational use (e.g. to remove clutter and permanent echoes), but we didn't further process the data. We contend that the effect of not conducting further processing would be negligible.

15. L119 and 120: "in this paper" repeated twice

This has now been revised.

16. L162 and elsewhere: it might be good to use the terminology "air temperature" instead of "temperature".

Noted. These have been revised.

17. L165: section title seems wrong here. The title has now been replaced by 'Vorticity-driven Lateral fire spread'.

18. Figure2: Wind direction? This is an important factor for fire behaviour and could help discuss the VLS aspects presented by the authors as one of the main mechanism for the increase in fire intensity? I personally prefer multiple subpanels as there is a bit of clutter on that figure with all the variables.

Wind direction has now been included, and plots separated into two: one for wind variables, the other one for the other weather variables. Interpretation of wind direction has been provided along with the figure as follows:

"Winds were majorly north-westerly, with no noticeable change in wind direction during the pyroCb period. However, a southerly change is observed at around 00:00 LT on 5 January, well after the Forcett-Dunalley pyroCb had already dissipated".

19. L224: "1km" above sea level? Yes. It now reads: "plume height from radar scans gradually increased from around 1 km above sea level at 13:00 to 8 km at around 15:00 and rapidly rose to the maximum injection height of 15 km…."

20. Figure 3: "Smoke on subpanel (a)" is not technically correct; rather "smoke plume" should be used. That is because radar at C-Band can't get good returns from smoke as such, as the authors correctly mentioned earlier in the draft; see McCarthy et al. (2018).

The legend in the subpanel (a) has now been corrected to reflect that it is a smoke plume.

21. L249 How is plume length defined?

We have included the description as follows: "Plume length is the horizontal distance between the origin of the plume and the farthest extent of the plume".

22. Figure 5, nice. Thanks. However, other reviewers have suggested changes regarding stratifying the fires into different fire size classes, to which we have slightly modified. It is now Figure 6.

23. Maybe recall for readers unfamiliar with Tasmanian seasons when that is?

The months in the fire season have now been included in the text. That information (including for non-fire season) is repeated in Fig. 7 captioning.

24. L315 The authors should cite Terrasson et al. here

The article has now been cited.

25. L322 325 The authors should cite Terrasson et al. here as well

The article has now been cited.

26. L338. The authors should discuss the findings of Terrasson et al. – the change of moisture in the lower and upper levels as brought by the cold front on the development of PyroCb and fire behaviour.

It has now been included as follows:

"… a number of Australian pyroCb events exhibit a distinct lack of midlevel moisture in their associated atmospheric profiles (*e.g.* the 2003 Canberra fire (Fromm et al., 2006) and the 2013 Wambelong fire (Wagga Wagga sounding for 13 January 2013 on http://weather.uwyo.edu/upperair/sounding.html). However, in this study, the mid- to upper-level moisture was higher during the time preceding pyroCb formation on 4 January (Fig. S2a), with a total precipitable water of 23 mm indicating a moist lower atmosphere (Webb and Fox-Hughes,

2015). Terrasson et al. (2019) report on the effect of a change in moisture between the low- and upper-levels (brought by a cold front) on the development of a pyroCb and enhancement of fire behaviour in the Sir Ivan fire in eastern Australia. Further research is required to properly understand the potential influence of mid-tropospheric moisture in driving pyroCb development in Australia".

27. L348: The authors should cite Terrasson et al. here
The article has now been cited.

28. L392, 394: The effect of the potential VLS on plume development is too speculative. Overall, I think that there is no justification based on the observations as shown in the current paper, for any effect of VLS.
We agree that the VLS-prone areas seem small, and their influence on pyroCb occurrence likely negligible based on these results, but we also caution that VLS likely occurred, but the effect may have been undermined by the limitations of the input data (DEM and wind direction layer).

We have now rephrased the sentence in both the results and discussion sections to read:
In section 3.1.3 (Results): "Evidence of an effect of Vorticity-driven Lateral Spread on the fire behaviour was not strong. Analysis of the precursor terrain conditions only revealed small patches of VLS-prone areas near Dunalley township (Fig. S4). However, we are not able to rule out VLS occurrence on parts of the terrain that were not resolved by the DEM, and which may have played a part in the evolution of the plume. Indeed, the lateral development of the upwind edge of the plume in Fig. S4 suggests lateral development of the fire, similar to that associated with VLS in other fires (McRae et al. 2015)."
In the discussion: "this study did not find strong evidence of the effect of VLS on fire behaviour, as indicated by small patches of VLS-prone areas near Dunalley township (Figure S4). It is therefore possible that the pyroCb attained its maximum height without VLS. However, this interpretation should be taken with caution as VLS possibly occurred but data constraints (especially the spatial resolution of the DEM and wind direction) may have precluded accurate determination of VLS".

29. The authors should cite Terrasson et al. here
The article has now been cited.

30. In Terrasson et al. (2019) we did exactly that, e.g. linking fire behaviour to plume development. The authors could also cite McCarthy et al. (2018) where fire behaviour is studied against plume development for a PyroCu in Victoria (Mt Bolton fire). McCarthy, N., H. McGowan, A. Guyot, and A. Dowdy, 2018: Mobile X-Pol Radar : A New Tool for Investigating Pyroconvection and Associated Wildfire Meteorology. Bull. Amer. Meteor. Soc., 99, 1177–1195, https://doi.org/10.1175/BAMS-D-16-0118.1

We have included the below sentence in the weather radar paragraph of the discussion:
"A feature of our study was linking plume evolution to fire severity mapping - an approach that has received limited attention. Duff et al. (2018) conducted one of the few studies, using statistical models to link the radar-detected plume volume to fire growth, and found that radar return volume (above a threshold of 10 dBZ) was a robust predictor of fire-area change. … [sentence removed] ……… Other Australian studies linking fire behaviour to radar-detected plume development include McCarthy et al. (2018) and Terrasson et al. (2019)".

31. Fig.S3: "the orange colour represents the most intense parts of the PyroCb": it would be appropriate to use a more scientific and precise language, i.e. provide the values or range of values for the equivalent reflectivity in dBZ (colour bar), and refer to the text of the article to describe how these have to be interpreted.

The colour bar has now been included in the maps (now as Figure 4), and the figure captioning updated to reflect the reflectivity values for the intense parts of the plume. We have now added a sentence in Section 3.1.3 of the main manuscript that describes pyroCb development in the atmosphere.

It reads:

"The period of the pyroCb in Fig. 4 is defined by very high radar returns, with reflectivity values of 48-88 dBZ, representing the most intense parts of the pyroCb. This strong reflectivity is indicative of high quantities of ash, and larger-sized hydrometeors such as ice crystals in the higher elevations".

Thanks and again I enjoyed reading your study. Kind regards. Thanks.

**Short comment- Anonymous**

This paper provides an analysis of a significant pyroCb event in Tasmania (January 2013). The evolution of the plume is assessed using weather radar. Analysis of meteorology is largely based on the lower troposphere using the C-Haines index and FFDI. Fire severity maps are also included. The manuscript is generally well-organized and easy to read. Results are relevant to several research communities. The pyroCb phenomenon has been gaining significant attention in recent years. However, a few aspects of the paper require some clarification prior to publication.

Please see my comments below.

Abstract line 18: "highly unstable atmosphere" do you mean "highly unstable lower troposphere" since you are referencing C-Haines?

We meant highly unstable lower atmosphere (the lower troposphere (C-Haines) and the low-mid- tropospheric instability (850-500 hPa in Fig. S2). We have now rephrased the sentence to: "The pyroCb was associated with a highly unstable lower atmosphere (C-Haines 10-11) …"

Lines 65-70: lapse rate in lower-troposphere, mid-levels, or both? The C-Haines is calculated from two lower-tropospheric levels. We have now rephrased the sentence as follows:

"The C-Haines index provides a measure of the potential for erratic fire behaviour, based on air temperature lapse and moisture content between two lower-tropospheric levels, and typically ranges from 0-13, although values above 13 are possible (Yeo et al., 2015; Di Virgilio et al., 2019)".

Line 139: "10 hPa". . ..what are these high altitude data used for? C-Haines is only based on the lower troposphere?

We were just indicating up to which atmospheric height the BARRA data were available, although we calculated up to the 150 hPa level (see the BARRA soundings in Fig. S2). We also, in Fig. S3, determined air temperature lapse rate at the 850-500 hPa, in addition to C-Haines (for 850-700 hPa height).

We have rephrased the sentence to: "We extracted hourly air temperature and dewpoint temperature at different air pressure levels (1000 hPa at the surface to 150 hPa in the lower stratosphere), as well as pre-calculated hourly McArthur FFDI for the period of the fire and the period of BARRA data available at the time of the study (January 2007-October 2016)".

Line 181: is there a reference for "typically happens at or near the start of fires"?

We have included an example of a fire whose major run was at the start of the fire. The sentence now reads:

"A total of 77 fires, of varying ignition sources, were identified in the Tasmania Fire Service fire history database as being >500 ha in size, and having a known ignition date. Of these, 18 did not have recorded end dates, and these were operationally specified as being four weeks later, an arbitrary cut-off to capture the most likely major growth (or 'run') of the fire that typically happens at or near the start of fires (e.g. the 2009 Victorian fires; Cruz et al., 2012). The assumption is that these runs result in peak fire intensities that most likely drive strong convection".

Line 195: please replace "was always higher than" with a specific range of values.

The sentence now reads: "It is worth noting that maximum daily FFDI calculated at the Hobart Airport station was always higher than daily gridded FFDI extracted from the BARRA product (compare maximum FFDI of 92 recorded at the station with FFDI of 68.7 from the BARRA model on 4 January)".

Section 3.1: In addition to surface fire weather, pyroCb are driven by the full 3-D structure of the troposphere. This discussion can be improved. The supplement provides thermodynamic profiles. Can you quantify the LCL and highlight the pressure levels that are used to calculate C-Haines? Was there any CAPE in these profiles? Please also provide a couple sentences describing the synoptic weather pattern that set the stage for this event.

We agree with your observations that both surface fire weather and 3D troposphere influence pyroCb development. We have captured the role of these fire weather aspects in the introduction and discussion sections. In Section 3.1 in the results, the atmospheric aspect has been described in the section on pyroCb development in the atmosphere, and the section on spatio-temporal variation of C-Haines. We have now included in section 3.1.2 information about synoptic weather as per your suggestions, to further expound on the tropospheric processes associated with the event.

- Re quantifying the Lifted Condensation Level, the values are now in the aerological profiles in Fig. S2 in the supplement.
- The pressure levels used to calculate C-Haines are 850 and 700 hPa, and are shown in Equations 1-3. Full description of the variables are provided immediately below the equations.
- CAPE is now available in the profiles. The value was zero, suggesting lower potential for thunderstorm development, with the energy to lift the air parcel and produce the pyroCb coming from the fire rather than the unmodified atmosphere. That information is provided in the supporting text in the supplement (section S1.2) as follows:
"The Convective Available Potential Energy (CAPE) as well as the maximum unstable CAPE (MUCAPE) during this period were zero, suggesting a lower potential for thunderstorm development. It is likely that much of the energy that lifted the air parcel and produced the pyroCb came from the fire itself rather than the unmodified atmosphere".

Section 3.1.2: Please show a basic radar image/map at the peak of the pyroCb event for reader orientation.

The 3D radar image has now been moved from the supplement to that section in the main article. It is now Figure 4.

We have added the following text: "The 850-500 hPa lapse rate gives an indication of the (in)stability of the lower half of the troposphere (1.3-5.5 km above sea level), with lapse rates of >7.5 °C km$^{-1}$ considered as very unstable lower atmosphere (Peterson et al., 2014)".

We have replaced "good correlation" with "moderate correlation". The sentence now reads: "…for the entire month of January 2013, there was a statistically significant moderate correlation (r = 0.5, $p$ <0.05) between these fire weather indices, although FFDI lagged C-Haines by around a day".

Yes, the wind change was as a result of a pre-frontal trough that passed before 00:00 on 5 January, well after the pyroCb had already established and dissipated. We have also included information on the synoptic weather in the section (3.1.2) preceding this section to provide context.

The sentence now reads:
"In the Tasman Peninsula, especially, C-Haines was mostly in the range of 10-12 for both days (peaking at 12-14) but moderated to 4-6 on 5 January after a southwest wind change at around 00:00, as a result of a pre-frontal trough crossing south-eastern Tasmania".

We greatly appreciate the reviewer's comments about making further use of the radar data. However, we feel that including all of the work suggested by the reviewer would go well beyond the intended scope of the manuscript, which was to provide a qualitative description of the evolution of the Forcett-Dunalley fire and pyroCb, and their relation to coincident surface weather and lower atmospheric conditions. Indeed, we feel that it would be more appropriate to pursue the work suggested by the reviewer in a separate and more detailed quantitative analysis, drawing upon the work of
Terrasson et al. (2019) and McCarthy et al. (2019). We have amended the discussion to specifically note this as a natural extension of the qualitative analysis presented in the manuscript.

The section in the discussion now reads: "… radars remain a reliable data source that can provide near-real time monitoring of strong pyroconvection, as evidenced by previous pyroCb studies in Australia and globally (Rosenfeld et al., 2007; Fromm et al., 2012; Lareau and Clements, 2016; Dowdy
et al., 2017; Peace et al., 2017; Lareau et al., 2018; Terrasson et al., 2019). This study did not analyse radial velocity from the Doppler radar; therefore future research on the Forcett-Dunalley fire and other fires should consider using that information to provide a more quantitative analysis of the thunderstorm, drawing upon previous work in Australia (McCarthy et al., 2019; Terrasson et al., 2019). A feature of our study was linking plume evolution to fire severity mapping - an approach that has received limited attention".

Lines 335-345: The soundings in the supplement do show the inverted-V structure typical of high-based convection, with some mid-level moisture in the profile at 1500. You can calculate total precipitable water to confirm. I believe that Peterson et al. provide more info on this.

The information on total precipitable water (and other metrics) is now provided in the profiles and in the discussion. The sentence in the discussion now reads:

"… a number of Australian pyroCb events exhibit a distinct lack of midlevel moisture in their associated atmospheric profiles (*e.g.* the 2003 Canberra fire (Fromm et al., 2006) and the 2013 Wambelong fire (Wagga Wagga sounding for 13 January 2013 on http://weather.uwyo.edu/upperair/sounding.html). However, in this study, the mid- to upper-level moisture was higher during the time preceding pyroCb formation on 4 January (Fig. S2a), with a total precipitable water of 23 mm indicating a moist lower atmosphere (Webb and Fox-Hughes, 2015). Terrasson et al. (2019) report on the effect of a change in moisture between the low- and upper-levels (brought by a cold front) on the development of a pyroCb and enhancement of fire behaviour in the Sir Ivan fire in eastern Australia. Further research is required to properly understand the potential influence of mid-tropospheric moisture in driving pyroCb development in Australia".

Supplement: Please annotate the time of the pyroCb on the relevant figures.

The time of the pyroCb are now shown in the figures, and are explained in the Fig. S3 captioning (now Fig. 4 in main article) as follows:

"Areas prone to vorticity-driven lateral spread (VLS, in red) overlaid with a 3D rendering of the vertical cross-section of plumes from the radar reflectivity for specific times during peak fire behaviour. Higher reflectivity values (dBZ >48) in all maps represent the most intense parts of the plume (and the pyroCb from 15:24 to 15:48 LT maps). The asterisk in the 15:24 map represents the likely initiation period of the pyroCb. Dunalley township is represented by a white star. A malfunction in radar for the 15:00 and 15:36 timestamps resulted in plume information to be only available at the lowest elevation angle of the radar scan, and are therefore not shown".

Supplement: Define/explain "BARRA pseudo-soundings"

Pseudo-soundings are atmospheric soundings obtained from the air and dewpoint temperatures produced by the BARRA model. They are not actual weather balloon (radiosonde) observations. We have now clarified in the supplement that the pseudo-soundings are vertical profiles sampled from the reanalysis data.

It would be nice to include a surface and upper level synoptic weather map that coincides with the soundings in Figure S1. This will provide some context on the large-scale meteorology driving this event.

We have now provided the synoptic charts in the supplement, and information on the synoptic weather on 3-4 January in Section 3.1.2 of the main article, which reads as follows:

On 3 January, a combination of: (1) a high-pressure system to the northeast of Tasmania and (2) a cold front and pre-frontal trough approaching from the west directed a freshening dry and hot northerly airstream over the island, favourable conditions for elevated fire danger (Bureau of Meteorology, 2013). By 08:00 LT on 4 January, the high-pressure system had moved only slowly eastward while the trough had progressed closer to western Tasmania (Fig. S1). This period coincided with moderate north-westerly winds in most locations in the state, except for the southeast (general area surrounding Dunalley) which recorded stronger winds and elevated fire danger. Fire danger steadily increased towards midday (the start of the first fire progression isochrone in Fig. 3c) and by 15:00, the leading edge of the trough was much closer to the west of Tasmania. An increase in pressure gradient brought about gusty conditions and catastrophic fire danger in some locations in southeast Tasmania, causing erratic fire behaviour on the Forcett-Dunalley fire. During that time, the trough crossed western Tasmania. By 17:00, when the pyroCb had likely dissipated (Fig. 3a), the pre-frontal trough was crossing Tasmania and fire danger subsequently reduced due to decreasing temperatures and winds. The trough continued to move eastwards and crossed southeast Tasmania after 23:00, leading to a west to southwest wind change by 00:00 on 5 January. The
front passed over the state early morning of 5 January and caused lightning and limited showers across Tasmania. Detailed analysis of the synoptic
weather patterns driving this event are provided in Bureau of Meteorology (2013).

Are the times listed local time or UTC? They are in LT. This information has now been updated in the relevant sections in the supplement.

if $(T_{850} - DT_{850}) > 30$, then $(T_{850} - DT_{850}) = 30$

if $CB > 5$, then $CB = 5 + (CB-5)/2$

CH = CA + CB                                                            (3)

where CA is a temperature lapse term; $T_{850}$ is air temperature at 850 hPa (or 1.3 km) atmospheric height; $T_{700}$ is air temperature at 700 hPa (around 3 km) height; $DT_{850}$ is the dewpoint temperature at 850 hPa height; CB is a dewpoint depression term; and CH is the continuous Haines Index (or C-Haines).

**2.2.3 Vorticity-driven Lateral fire spread**

We also determined whether the period of rapid cloud/plume development coincided with local surface dynamics, which likely enhanced fire behaviour. Specifically, the effect of Vorticity-driven Lateral Spread (VLS) on fire behaviour was tested. VLS is an atypical fire spread arising from the interaction between strong winds and terrain which creates lee-slope eddies that
interact with the fire to cause lateral fire propagation, an increase in fire intensity and mass spotting downwind of the lateral spread zone (Sharples et al., 2012). VLS-prone areas were defined according to Sharples et al. (2012) criteria as follows: lee-facing slopes steeper than 15°, with slopes facing to approximately 40° of the direction the wind is blowing towards. A wind direction layer (mostly north-westerly) at 16:00 on 4 January (the period around peak plume height) was extracted from the BARRA dataset and resampled to correspond to the spatial resolution of the Digital Elevation Model (33 m), provided by the
Tasmanian Department of Primary Industries, Parks, Water and Environment. Both layers were combined using the aforementioned criteria, resulting in a binary map where areas fulfilling the criteria of VLS were assigned a value of one and all other areas zero.

**2.4 Spatio-temporal context of fire weather in Tasmania**

### 2.4.1 Weather conditions during large Tasmanian fires

We compared FFDI and C-Haines values during the Forcett-Dunalley fire and values associated with other large Tasmanian fires between years 2007 and 2016 (the period of available BARRA data). A total of 77 fires, of varying ignition sources, were identified in the Tasmania Fire Service fire history database as being >500 ha in size, and having a known ignition date. Of these, 18 did not have recorded end dates, and these were operationally specified as being four weeks later, an arbitrary cut-
off to capture the most likely major growth (or 'run') of the fire that typically happens at or near the start of fires (e.g. the 2009 Victorian fires; Cruz et al., 2012). The assumption is that these runs result in peak fire intensities that most likely drive strong convections. We subsequently produced a scatterplot of highest daily FFDI for the duration of each of the fires and the associated C-Haines value. These data were overlaid on a density map of C-Haines and FFDI values for all cells (and days) in Tasmania during the entire BARRA period to provide background weather conditions for Tasmania.

**2.4.2 Elevated fire weather in Tasmania**

To provide a geographic context to the Forcett-Dunalley fire, the spatial patterns of days conducive for extreme fire behaviour in Tasmania were mapped by determining counts of days that exceeded combined C-Haines and FFDI thresholds of 9 and 25 respectively, then aggregating them to fire season (October-March) and non-fire season (April-September). These thresholds were defined to correspond to an elevated fire weather day (95th percentile) based on weather conditions at Hobart Airport, which represents the broader airmass in south-eastern Tasmania. For example, the FFDI threshold is close to the 95th percentile (FFDI 31) observed during the 1998-2005 fire seasons (Marsden-Smedley, 2014). We chose to use FFDI to describe near-surface fire weather across the state, but acknowledge that for some fuel types (*e.g.* moorlands) occurring in Tasmania (Marsden-Smedley and Catchpole, 1995), other indices such as the Grassland/Moorland Fire Danger Index may be more suitable. It is worth noting that maximum daily FFDI calculated at the Hobart Airport station was always higher than daily gridded FFDI extracted from the BARRA product (compare maximum FFDI of 92 recorded at the station with FFDI of 68.7 from the BARRA model on 4 January).

**3 Results**

**3.1 Fire weather during the fire**

**3.1.1 Surface fire weather**

Dangerous fire weather conditions in southeast Tasmania were observed from 3-4 January 2013. Hobart Airport, at 14:00 LT on 3 January when the fire was reported, recorded 33 $^{\circ}$C (the maximum temperature for the 3rd), relative humidity (RH) of 15 %, and strong north westerly winds reaching 37 km h$^{-1}$ and gusting to 55 km h$^{-1}$ (Fig. 2). The smoke plume was detectable by weather radar between 15:18 and 19:00 LT. The weather conditions deteriorated on 4 January, with extreme maximum temperatures (reaching 40 $^{\circ}$C), strong winds (35-46 km h$^{-1}$, gusts of 60-70 km h$^{-1}$), and low RH (11 %) in the afternoon. At 12:00, the Forcett-Dunalley plume was again detectable by weather radar (Figs. 2 & 3a), when values of the weather variables peaked and were maintained during the period of violent pyroconvection between 15:24-16:30 (Fig. 2). Winds were majorly north-westerly, with no noticeable change in wind direction during the pyroCb period. However, a southerly change is observed at around 00:00 LT on 5 January, well after the Forcett-Dunalley pyroCb had already dissipated.

[Figure]

**Figure 2:** Time series of 30-minute weather data obtained from 3-4 January 2013 at Hobart Airport. (a) Rainfall, Relative Humidity (RH) and air temperature. (b) Wind variables that include wind speed, wind gust and wind direction. Units: rainfall (mm), RH (%), air temperature (ºC), wind gust and speed (km h⁻¹), and wind direction (as cardinal direction). The arrows point in the direction wind is blowing to. The vertical lines represent the time of the start of the fire and the period of the pyroCb event. The periods of the radar detection of the plume for the two days are also shown on the graphs.

**3.1.2 Synoptic weather**

On 3 January, a combination of: (1) a high-pressure system to the northeast of Tasmania and (2) a cold front and pre-frontal trough approaching from the west directed a freshening dry and hot northerly airstream over the island; favourable conditions for elevated fire danger (Bureau of Meteorology, 2013). By 08:00 LT on 4 January, the high-pressure system had moved only slowly eastward while the trough had progressed closer to western Tasmania (Fig. S1). This period coincided with moderate north-westerly winds in most locations in the state, except for the southeast (general area surrounding Dunalley) which recorded stronger winds and elevated fire danger. Fire danger steadily increased towards midday (the start of the first fire progression isochrone in Fig. 3c) and by 15:00, the leading edge of the trough was much closer to the west of Tasmania. An increase in pressure gradient brought about gusty conditions and catastrophic fire danger in some locations in southeast Tasmania, causing erratic fire behaviour on the Forcett-Dunalley fire. During that time, the trough crossed western Tasmania. By 17:00, when the pyroCb had likely dissipated (Fig. 3a), the pre-frontal trough was crossing Tasmania and fire danger subsequently reduced due to decreasing temperatures and winds. The trough continued to move eastwards and crossed southeast Tasmania after 23:00, leading to a west to southwest wind change by 00:00 on 5 January. The front passed over the state early morning of 5 January and caused lightning and limited showers across Tasmania. Detailed analysis of the synoptic weather patterns driving this event are provided in Bureau of Meteorology (2013).

**3.1.3 PyroCb development in the atmosphere**

On 4 January, the plume height from radar scans gradually increased from around 1 km above sea level at 13:00 to 8 km at around 15:00 and rapidly rose to the maximum injection height of 15 km (lower stratosphere) between 15:24 and 15:48 (Fig. 3a), representing the peak of pyroconvection. During the period of violent pyroconvection, thunderstorms developed and moved in a south-easterly direction towards the Tasman Sea, causing two lightning strikes, which were detected around 16:10 (Bureau of Meteorology, 2013). Radar returns during the peak period (Fig. 4) were likely due to glaciation within the pyroCb and precipitation at high altitudes, which then evaporated before reaching the surface due to intense heat, while the cooled air mass from evaporation descended towards the surface as convective outflows, generating gusty and erratic surface winds. The cloud top then decreased to around 7 km at 16:42, after which the pyroCb likely dissipated and the plume subsequently stabilised at heights of 3 km. Atmospheric instability and the pyroCb dynamics are confirmed by atmospheric soundings from BARRA for 15:00-16:00 on 4 January (Fig. S2), and a time series of 850-500 hPa air temperature lapse rate on 4 January (Fig. S3). The soundings show air temperature at the tropopause (~12 km, below the pyroCb height) to be -50 $^{\circ}$C, supporting the observation of electrification of the pyroCb. Electrification/lightning typically occurs at temperatures around or below -20 $^{\circ}$C (Williams, 1989). The 850-500 hPa lapse rate gives an indication of the (in)stability of the lower half of the troposphere (1.3-5.5 km above sea level), with lapse rates of >7.5 $^{\circ}$C km$^{-1}$ considered as very unstable lower atmosphere (Peterson et al., 2014).

[Figure]

**Figure 3:** Plume dimensions, FFDI and fire severity traces during the evolution of a pyroCb on the afternoon of 4 January. (a) Six-minute variation of plume dimensions during peak fire behaviour, where plume height has been scaled for visualisation by multiplying by a factor of 10. The asterisk (*) represents the period (16:42-17:06) with missing weather radar data. (b) FFDI trace obtained from Hobart Airport weather station, and C-Haines computed from the BARRA product during the corresponding period of smoke plume growth. (c) Temporal pattern of fire severity, adapted from Ndalila et al. (2018). Black vertical lines in the three graphs represent the period of violent pyro-convection.

For the horizontal plume dimensions (length, area and perimeter), there was a lagged response to the effect of intense fire activity that occurred at around 15:24 LT. While a maximum in cloud height was observed at 15:48, other plume metrics peaked at around 17:13 (Fig. 3a). Plume length is defined as the horizontal distance between the origin of the plume and its farthest extent.

925 The period of drastic increase in the height of the pyroCb cloud was associated with elevated FFDI of 60-75 (Severe (50-74) to marginally Extreme (75) fire danger classes), elevated gridded C-Haines of 10-11.1 at Hobart Airport (Fig. 3b), a large area burnt (approx.10,000 ha) and the highest proportion of burning of the two highest fire severity (total crown defoliation) categories in dry *Eucalyptus* forests (Fig. 3c). The isochrone leading up to the peak in fire behaviour accounted for 10 % of total area burnt across all vegetation types within the entire fire perimeter, the peak period contributed 46 %, while the last

930 isochrone on 4 January contributed 9 % of total area burnt (Fig. 3c). Evidence of an effect of Vorticity-driven Lateral Spread on the fire behaviour was not strong. Analysis of the precursor terrain conditions only revealed small patches of VLS-prone areas near Dunalley township (Figs. 4 and S4). However, we are not able to rule out VLS occurrence on parts of the terrain that were not resolved by the DEM, and which may have played a part in the evolution of the plume. Indeed, the lateral development of the upwind edge of the plume in Fig. S4 suggests lateral development of the fire, similar to that associated

935 with VLS in other fires (McRae et al., 2015). The majority of the plume area (>70%) extended over the Tasman Sea in a south-easterly direction from the location of the fire, under the influence of above-surface winds (Fig. 4). The period of the pyroCb in Fig. 4 is defined by very high radar returns, with reflectivity values of 48-88 dBZ, representing the most intense parts of the pyroCb. This strong reflectivity is indicative of high quantities of ash, and larger-sized hydrometeors such as ice crystals in the higher elevations.

[Figure]

**Figure 4:** Areas prone to vorticity-driven lateral spread (VLS, in red) overlaid with a 3D rendering of the vertical cross-section of plumes from the radar reflectivity for specific times during peak fire behaviour. Higher reflectivity values (dBZ >48) in all maps represent the most intense parts of the plume (and the pyroCb which occurred from around 15:24 to 16:30 LT). The asterisk in Fig. 4b represents the likely initiation period of the pyroCb. Dunalley township is represented by a white star. A malfunction in radar for the 15:30 and 15:36 timestamps resulted in plume information to be only available at the lowest elevation angle of the radar scan and are therefore not shown.

**3.1.4 Spatio-temporal variation of C-Haines**

During the development of the pyroCb, maximum daily FFDI and gridded C-Haines on 4 January at Hobart Airport were consistently high (92 and 11.1, respectively). In the following days both these indices markedly declined (Fig. S5); indeed, for the entire month of January 2013, there was a statistically significant moderate correlation ($r = 0.5$, $p <0.05$) between these fire weather indices, although FFDI lagged C-Haines by around a day. At a state-wide scale, from 3 to 4 January, the whole of Tasmania displayed dangerous fire weather, particularly south-eastern Tasmania, which recorded high levels of C-Haines (Fig. 5). In the Tasman Peninsula, especially, C-Haines was mostly in the range of 10-12 for both days (peaking at 12-14) but moderated to 4-6 on 5 January after a southwest wind change at around 00:00, as a result of a pre-frontal trough crossing south-eastern Tasmania. A complete description of overall meteorological conditions, including the aerological diagrams, can be found in the supplementary section.

[Figure]

**Figure 5:** Spatio-temporal distribution of maximum daily C-Haines for Tasmania for 3-5 January 2013. The blue square indicates the location of the Tasman Peninsula.

**3.2 Contextualising the Forcett-Dunalley pyroCb**

**3.2.1 Fire weather in large Tasmanian fires**

[revised manuscript text omitted]

Unidata: Integrated Data Viewer (IDV) version 5.5, Boulder, CO: UCAR/Unidata, 2018

Webb, M., and Fox-Hughes, P.: An analysis of extreme rainfall in northern Tasmania, 12-14 January 2011. Bureau Research Report No. 003, Bureau of Meteorology, Hobart, 2015.

Williams, E. R.: The tripole structure of thunderstorms, J. Geophys. Res. Atmos., 94, 13151-13167, doi:10.1029/JD094iD11p13151, 1989.

Yeo, C. S., Kepert, J. D., and Hicks, R.: Fire danger indices: current limitations and a pathway to better indices., Bushfire & Natural Hazards CRC, Australia., 2015.

**Evolution of a **pyrocumulonimbus** event associated with an extreme wildfire in Tasmania, Australia**

[revised manuscript text omitted]

2007; Fromm et al., 2010; Lareau and Clements, 2016; Di Virgilio et al., 2019). A fire weather index commonly used in

Australia to monitor meteorological conditions in the lower atmosphere is the continuous Haines Index (C-Haines; Mills and McCaw, 2010). The index is a modification from the Haines index (Haines, 1988) which is routinely used in the US, but adapted to suit the frequent hot and dry summer conditions in Australia. The C-Haines index provides a measure of the potential for erratic fire behaviour, based on air temperature lapse and moisture content between two lower-tropospheric levels, and typically ranges from 0-13, although values above 13 are possible (Yeo et al., 2015; Di Virgilio et al., 2019). High values of C-Haines imply drier and more unstable atmospheric conditions, which favour lifting of the heated air higher into the atmosphere. In particular, a large air temperature lapse in the atmosphere favours the maintenance of strong convection and increases the likelihood of pyroCb development. The role of tropospheric temperature and moisture in pyroCb dynamics is exemplified in the inverted-V thermodynamic profile (Peterson et al., 2017). The profile shows a dry and warm near-surface environment in which temperature decreases adiabatically with altitude to the top of the mixed layer (~ 3 km) where relative humidity is higher. Altitudes immediately above the mixed layer are usually drier, and this dry air can mix to the surface in strong convective downdrafts, increasing surface fire behaviour (McRae et al., 2015). Further, higher mid-troposphere moisture can interact with weaker wind shear and high temperature lapse rate to produce strong convective updrafts (Peterson et al., 2017).

[revised manuscript text omitted]

to determine any congruence between pyroCb dynamics and fire weather, area burnt and fire severity patterns during the period of erratic fire behaviour. The chronosequence of fire severity was derived from a previous study (Ndalila et al., 2018) from the intersection of fire severity map with fire progression isochrones within the fireground. Fire severity was based on differential normalised burn ratio (dNBR;  Key and Benson, 2006) analysis of pre- and post-fire 30 m resolution Landsat 7 satellite images.

A detailed description of fire severity assessments is provided by Ndalila et al. (2018).

**2.2.2 C-Haines analysis**

We obtained gridded weather reanalysis data from the Bureau of Meteorology Atmospheric high-resolution Regional

Reanalysis for Australia (BARRA), downscaled for the Tasmanian sub-domain (BARRA-TA) to 1.5 km spatial resolution (Su et al., 2019). BARRA combines numerical weather forecasts with observational data to produce realistic depictions of surface meteorology and atmospheric conditions. We extracted hourly air temperature and dewpoint temperature at different air pressure levels (1000 hPa at the surface to 150 hPa in the lower stratosphere), as well as pre-calculated hourly McArthur FFDI

for the period of the fire and the period of BARRA data available at the time of the study (January 2007-October 2016).

Extraction, conversion and general BARRA analysis was performed in R version 3.4.0 software (R Core Team, 2017).

At each grid cell in Tasmania, the C-Haines index was calculated from the hourly estimates of air temperature and dewpoint temperature at relevant atmospheric levels (850 and 700 hPa) based on Eqs. 1-3. We preferred the BARRA product to radiosonde data to calculate C-Haines because of: (1) a possible geographic drift of the weather balloon as it rises through the atmosphere, resulting in inconsistencies in locations where data were recorded; (2) availability of balloon data only twice a day at a single location, which is unrepresentative of many regions; and (3) the BARRA product combines other data sources such as satellite observations to model air temperature and moisture. Nevertheless, we validated our C-Haines values by using the radiosonde data for Hobart Airport, establishing a correlation of 0.74 between the two datasets. Our calculated hourly C-

Haines values, as well as the extracted hourly FFDI for Tasmania were then aggregated to maximum daily values. Temporal maps of daily C-Haines distribution were then produced for the first three days of the fire, between 3-5 January.

$$CA = (T850 – T700)/2 − 2 \qquad\qquad (1)$$

$$CB = (T850 − DT850)/3 − 1 \qquad\qquad (2)$$

if $(T_{850} - DT_{850}) > 30$, then $(T_{850} - DT_{850}) = 30$

if $CB > 5$, then $CB = 5 + (CB-5)/2$

$$CH = CA + CB \tag{3}$$

where CA is a temperature lapse term; $T_{850}$ is air temperature at 850 hPa (or 1.3 km) atmospheric height; $T_{700}$ is air temperature at 700 hPa (around 3 km) height; $DT_{850}$ is the dewpoint temperature at 850 hPa height; CB is a dewpoint depression term; and CH is the continuous Haines Index (or C-Haines).

**2.2.3 Vorticity-driven Lateral fire spread**

We also determined whether the period of rapid cloud/plume development coincided with local surface dynamics, which likely enhanced fire behaviour. Specifically, the effect of Vorticity-driven Lateral Spread (VLS) on fire behaviour was tested. VLS is an atypical fire spread arising from the interaction between strong winds and terrain which creates lee-slope eddies that interact with the fire to cause lateral fire propagation, an increase in fire intensity and mass spotting downwind of the lateral spread zone (Sharples et al., 2012). VLS-prone areas were defined according to Sharples et al. (2012) criteria as follows: lee- facing slopes steeper than 15°, with slopes facing to approximately 40° of the direction the wind is blowing towards. A wind direction layer (mostly north-westerly) at 16:00 on 4 January (the period around peak plume height) was extracted from the BARRA dataset and resampled to correspond to the spatial resolution of the Digital Elevation Model (33 m), provided by the Tasmanian Department of Primary Industries, Parks, Water and Environment. Both layers were combined using the aforementioned criteria, resulting in a binary map where areas fulfilling the criteria of VLS were assigned a value of one and all other areas zero.

**2.4 Spatio-temporal context of fire weather in Tasmania**

**2.4.1 Weather conditions during large Tasmanian fires**

We compared FFDI and C-Haines values during the Forcett-Dunalley fire and values associated with other large Tasmanian fires between years 2007 and 2016 (the period of available BARRA data). A total of 77 fires, of varying ignition sources, were identified in the Tasmania Fire Service fire history database as being >500 ha in size, and having a known ignition date. Of these, 18 did not have recorded end dates, and these were operationally specified as being four weeks later, an arbitrary cut-off to capture the most likely major growth (or 'run') of the fire that typically happens at or near the start of fires (e.g. the 2009 Victorian fires; Cruz et al., 2012). The assumption is that these runs result in peak fire intensities that most likely drive strong convections. We subsequently produced a scatterplot of highest daily FFDI for the duration of each of the fires and the associated C-Haines value. These data were overlaid on a density map of C-Haines and FFDI values for all cells (and days) in Tasmania during the entire BARRA period to provide background weather conditions for Tasmania.

**2.4.2 Elevated fire weather in Tasmania**

To provide a geographic context to the Forcett-Dunalley fire, the spatial patterns of days conducive for extreme fire behaviour in Tasmania were mapped by determining counts of days that exceeded combined C-Haines and FFDI thresholds of 9 and 25 respectively, then aggregating them to fire season (October-March) and non-fire season (April-September). These thresholds were defined to correspond to an elevated fire weather day (95th percentile) based on weather conditions at Hobart Airport, which represents the broader airmass in south-eastern Tasmania. For example, the FFDI threshold is close to the 95th percentile (FFDI 31) observed during the 1998-2005 fire seasons (Marsden-Smedley, 2014). We chose to use FFDI to describe near-surface fire weather across the state, but acknowledge that for some fuel types (*e.g.* moorlands) occurring in Tasmania (Marsden-Smedley and Catchpole, 1995), other indices such as the Grassland/Moorland Fire Danger Index may be more suitable. It is worth noting that maximum daily FFDI calculated at the Hobart Airport station was always higher than daily gridded FFDI extracted from the BARRA product (compare maximum FFDI of 92 recorded at the station with FFDI of 68.7

from the BARRA model on 4 January).

**3 Results**

**3.1 Fire weather during the fire**

**3.1.1 Surface fire weather**

Dangerous fire weather conditions in southeast Tasmania were observed from 3-4 January 2013. Hobart Airport, at 14:00 LT

on 3 January when the fire was reported, recorded 33 °C (the maximum temperature for the 3rd), relative humidity (RH) of 15 %, and strong north westerly winds reaching 37 km h$^{-1}$ and gusting to 55 km h$^{-1}$ (Fig. 2). The smoke plume was detectable by weather radar between 15:18 and 19:00 LT. The weather conditions deteriorated on 4 January, with extreme maximum temperatures (reaching 40 °C), strong winds (35-46 km h$^{-1}$, gusts of 60-70 km h$^{-1}$), and low RH (11 %) in the afternoon. At 12:00, the Forcett-Dunalley plume was again detectable by weather radar (Figs. 2 & 3a), when values of the weather variables peaked and were maintained during the period of violent pyroconvection between 15:24-16:30 (Fig. 2). Winds were majorly north-westerly, with no noticeable change in wind direction during the pyroCb period. However, a southerly change is observed at around 00:00 LT on 5 January, well after the Forcett-Dunalley pyroCb had already dissipated.

[Figure]

**Figure 2:** Time series of 30-minute weather data obtained from 3-4 January 2013 at Hobart Airport. (a) Rainfall, Relative Humidity (RH) and air temperature. (b) Wind variables that include wind speed, wind gust and wind direction. Units: rainfall (mm), RH (%), air temperature (°C), wind gust and speed (km h⁻¹), and wind direction (as cardinal direction). The arrows point in the direction wind is blowing to. The vertical lines represent the time of the start of the fire and the period of the pyroCb event. The periods of the radar detection of the plume for the two days are also shown on the graphs.

**3.1.2 Synoptic weather**

On 3 January, a combination of: (1) a high-pressure system to the northeast of Tasmania and (2) a cold front and pre-frontal trough approaching from the west directed a freshening dry and hot northerly airstream over the island; favourable conditions for elevated fire danger (Bureau of Meteorology, 2013). By 08:00 LT on 4 January, the high-pressure system had moved only slowly eastward while the trough had progressed closer to western Tasmania (Fig. S1). This period coincided with moderate north-westerly winds in most locations in the state, except for the southeast (general area surrounding Dunalley) which recorded stronger winds and elevated fire danger. Fire danger steadily increased towards midday (the start of the first fire progression isochrone in Fig. 3c) and by 15:00, the leading edge of the trough was much closer to the west of Tasmania. An increase in pressure gradient brought about gusty conditions and catastrophic fire danger in some locations in southeast Tasmania, causing erratic fire behaviour on the Forcett-Dunalley fire. During that time, the trough crossed western Tasmania. By 17:00, when the pyroCb had likely dissipated (Fig. 3a), the pre-frontal trough was crossing Tasmania and fire danger subsequently reduced due to decreasing temperatures and winds. The trough continued to move eastwards and crossed southeast Tasmania after 23:00, leading to a west to southwest wind change by 00:00 on 5 January. The front passed over the state early morning of 5 January and caused lightning and limited showers across Tasmania. Detailed analysis of the synoptic weather patterns driving this event are provided in Bureau of Meteorology (2013).

**3.1.3 PyroCb development in the atmosphere**

On 4 January, the plume height from radar scans gradually increased from around 1 km above sea level at 13:00 to 8 km at around 15:00 and rapidly rose to the maximum injection height of 15 km (lower stratosphere) between 15:24 and 15:48 (Fig. 3a), representing the peak of pyroconvection. During the period of violent pyroconvection, thunderstorms developed and moved in a south-easterly direction towards the Tasman Sea, causing two lightning strikes, which were detected around 16:10 (Bureau of Meteorology, 2013). Radar returns during the peak period (Fig. 4) were likely due to glaciation within the pyroCb and precipitation at high altitudes, which then evaporated before reaching the surface due to intense heat, while the cooled air mass from evaporation descended towards the surface as convective outflows, generating gusty and erratic surface winds. The cloud top then decreased to around 7 km at 16:42, after which the pyroCb likely dissipated and the plume subsequently stabilised at heights of 3 km. Atmospheric instability and the pyroCb dynamics are confirmed by atmospheric soundings from BARRA for 15:00-16:00 on 4 January (Fig. S2), and a time series of 850-500 hPa air temperature lapse rate on 4 January (Fig. S3). The soundings show air temperature at the tropopause (~12 km, below the pyroCb height) to be -50 $^{\circ}$C, supporting the observation of electrification of the pyroCb. Electrification/lightning typically occurs at temperatures around or below -20 $^{\circ}$C (Williams, 1989). The 850-500 hPa lapse rate gives an indication of the (in)stability of the lower half of the troposphere (1.3-5.5 km above sea level), with lapse rates of >7.5 $^{\circ}$C km$^{-1}$ considered as very unstable lower atmosphere (Peterson et al., 2014).

[Figure]

**Figure 3:** Plume dimensions, FFDI and fire severity traces during the evolution of a pyroCb on the afternoon of 4 January. (a) Six-minute variation of plume dimensions during peak fire behaviour, where plume height has been scaled for visualisation by multiplying by a factor of 10. The asterisk (*) represents the period (16:42-17:06) with missing weather radar data. (b) FFDI trace obtained from Hobart Airport weather station, and C-Haines computed from the BARRA product during the corresponding period of smoke plume growth. (c) Temporal pattern of fire severity, adapted from Ndalila et al. (2018). Black vertical lines in the three graphs represent the period of violent pyro-convection.

For the horizontal plume dimensions (length, area and perimeter), there was a lagged response to the effect of intense fire activity that occurred at around 15:24 LT. While a maximum in cloud height was observed at 15:48, other plume metrics peaked at around 17:13 (Fig. 3a). Plume length is defined as the horizontal distance between the origin of the plume and its farthest extent.

The period of drastic increase in the height of the pyroCb cloud was associated with elevated FFDI of 60-75 (Severe (50-74) to marginally Extreme (75) fire danger classes), elevated gridded C-Haines of 10-11.1 at Hobart Airport (Fig. 3b), a large area burnt (approx.10,000 ha) and the highest proportion of burning of the two highest fire severity (total crown defoliation) categories in dry *Eucalyptus* forests (Fig. 3c). The isochrone leading up to the peak in fire behaviour accounted for 10 % of total area burnt across all vegetation types within the entire fire perimeter, the peak period contributed 46 %, while the last isochrone on 4 January contributed 9 % of total area burnt (Fig. 3c). Evidence of an effect of Vorticity-driven Lateral Spread on the fire behaviour was not strong. Analysis of the precursor terrain conditions only revealed small patches of VLS-prone areas near Dunalley township (Figs. 4 and S4). However, we are not able to rule out VLS occurrence on parts of the terrain that were not resolved by the DEM, and which may have played a part in the evolution of the plume. Indeed, the lateral development of the upwind edge of the plume in Fig. S4 suggests lateral development of the fire, similar to that associated with VLS in other fires (McRae et al., 2015). The majority of the plume area (>70%) extended over the Tasman Sea in a south-easterly direction from the location of the fire, under the influence of above-surface winds (Fig. 4). The period of the pyroCb in Fig. 4 is defined by very high radar returns, with reflectivity values of 48-88 dBZ, representing the most intense parts of the pyroCb. This strong reflectivity is indicative of high quantities of ash, and larger-sized hydrometeors such as ice crystals in the higher elevations.

[Figure]

**Figure 4:** Areas prone to vorticity-driven lateral spread (VLS, in red) overlaid with a 3D rendering of the vertical cross-section of plumes from the radar reflectivity for specific times during peak fire behaviour. Higher reflectivity values (dBZ >48) in all maps represent the most intense parts of the plume (and the pyroCb which occurred from around 15:24 to 16:30 LT). The asterisk in Fig. 4b represents the likely initiation period of the pyroCb. Dunalley township is represented by a white star. A malfunction in radar for the 15:30 and 15:36 timestamps resulted in plume information to be only available at the lowest elevation angle of the radar scan and are therefore not shown.

**3.1.4 Spatio-temporal variation of C-Haines**

During the development of the pyroCb, maximum daily FFDI and gridded C-Haines on 4 January at Hobart Airport were consistently high (92 and 11.1, respectively). In the following days both these indices markedly declined (Fig. S5); indeed, for the entire month of January 2013, there was a statistically significant moderate correlation ($r = 0.5$, $p < 0.05$) between these fire weather indices, although FFDI lagged C-Haines by around a day. At a state-wide scale, from 3 to 4 January, the whole of Tasmania displayed dangerous fire weather, particularly south-eastern Tasmania, which recorded high levels of C-Haines (Fig. 5). In the Tasman Peninsula, especially, C-Haines was mostly in the range of 10-12 for both days (peaking at 12-14) but moderated to 4-6 on 5 January after a southwest wind change at around 00:00, as a result of a pre-frontal trough crossing south-eastern Tasmania. A complete description of overall meteorological conditions, including the aerological diagrams, can be found in the supplementary section.

[Figure]

**Figure 5:** Spatio-temporal distribution of maximum daily C-Haines for Tasmania for 3-5 January 2013. The blue square indicates the location of the Tasman Peninsula.

**3.2 Contextualising the Forcett-Dunalley pyroCb**

**3.2.1 Fire weather in large Tasmanian fires**

[revised manuscript text omitted]

Turbulence associated with winds and terrain has been suggested as an important factor contributing to pyroCb formation via amplification of fire through processes such as mass spotting, topographic channelling of winds and vorticity-driven lateral spread (VLS) (Sharples et al., 2012). Previous studies have linked abrupt increases in plume height with VLS; these include the 2003 Canberra fires (Sharples et al., 2012) and the 2006 Grose Valley fire (McRae et al., 2015). VLS, in those studies, was confirmed based on the observation of rapid lateral expansion of the plume (McRae, 2010). However, this study did not find strong evidence of the effect of VLS on fire behaviour, as indicated by small patches of VLS-prone areas near Dunalley township (Fig. S4). It is therefore possible that the pyroCb attained its maximum height without the influence of VLS. However, this interpretation should be taken with caution as VLS possibly occurred but data constraints (especially the spatial resolution of the DEM and wind direction) may have precluded accurate determination of VLS. Mapping of the Forcett-Dunalley fire by Ndalila et al. (2018) showed that areas subjected to the highest fire intensities broadly aligned with undulating terrain and long unburnt dry *Eucalyptus* forest which under the influence of strong winds produced an ember storm that impacted the coastal township of Dunalley situated in the lee of the low hills. During this period (at 15:25 on 4 January), the rate of fire spread was reported to be around 50 m min$^{-1}$ (or 3 km h$^{-1}$), which then reduced to 1.9 km h$^{-1}$ between 17:30 and 20:00, and by the time of the next fire isochrone at 22:00, when fire severity had significantly reduced (Fig. 3), the rate of spread was 1 km h$^{-1}$ (Marsden-Smedley, 2014). It must be acknowledged that the role of downdrafts and mass spotting underneath the plume during the period of extreme fire behaviour is hard to infer without additional data sources on fire behaviour (such as infra-red/multispectral linescans) and high-resolution coupled fire-atmosphere modelling (Peace et al., 2015).

This study hinges on the application of weather radars to track the evolution of a pyroCb. It is worth noting that weather radars are not perfectly suited for all fires because of their limited geographic range (relative to satellite observations) and their inability to detect microscale cloud particles (<100 µm). Nonetheless, radars remain a reliable data source that can provide near-real time monitoring of strong pyroconvection, as evidenced by previous pyroCb studies in Australia and globally (Rosenfeld et al., 2007; Fromm et al., 2012; Lareau and Clements, 2016; Dowdy et al., 2017; Peace et al., 2017; Lareau et al., 2018; Terrasson et al., 2019). This study did not analyse radial velocity from the Doppler radar; therefore future research on the Forcett-Dunalley fire and other fires should consider using that information to provide a more quantitative analysis of the thunderstorm, drawing upon previous work in Australia (McCarthy et al., 2019; Terrasson et al., 2019). A feature of our study was linking plume evolution to fire severity mapping - an approach that has received limited attention. Duff et al. (2018) conducted one of the few studies, using statistical models to link the radar-detected plume volume to fire growth, and found that radar return volume (above a threshold of 10 dBZ) was a robust predictor of fire-area change. These results appear consistent with the present findings, particularly with the correlation between rapid plume development and the horizontal growth of the fire. Other Australian studies linking fire behaviour to radar-detected plume development include McCarthy et al. (2018) and Terrasson et al. (2019).

[revised manuscript text omitted]

Unidata: Integrated Data Viewer (IDV) version 5.5, Boulder, CO: UCAR/Unidata, 2018

Webb, M., and Fox-Hughes, P.: An analysis of extreme rainfall in northern Tasmania, 12-14 January 2011. Bureau Research Report No. 003, Bureau of Meteorology, Hobart, 2015.

Williams, E. R.: The tripole structure of thunderstorms, J. Geophys. Res. Atmos., 94, 13151-13167, doi:10.1029/JD094iD11p13151, 1989.

Yeo, C. S., Kepert, J. D., and Hicks, R.: Fire danger indices: current limitations and a pathway to better indices., Bushfire & Natural Hazards CRC, Australia., 2015.

**Evolution of a pyrocumulonimbus event associated with an extreme wildfire in Tasmania, Australia**

**Supplements**

**S1 Atmospheric profile during the fire**

**S1.1 Synoptic weather associated with the fire**

See text in Section 3.1.2 of the main article for details.

[Figure]

Fig. S1: Mean sea level pressure charts for 3-4 January, adopted from Bureau of Meteorology (2013). (a) represents synoptic conditions at 11:00 LT on 3 January, while b-d represent conditions at 05:00, 11:00 and 17:00 LT on 4 January 2013.

**S1.2 BARRA soundings**

In the BARRA pseudo-soundings (vertical profiles sampled from the reanalysis data) on 4 January, the atmosphere was unstable, as indicated by a sharp decrease in air temperature with height through the troposphere (Figs. S2a-b). The tropopause (often evident as a strong inversion) was not clearly defined, at least to approximately 200 hPa (11.7 km), thus contributing to the vertical development of the pyroCb. It is worth noting that the mid- to upper-level moisture in the figures is higher, suggesting cloud formation near 400 hPa (7.1 km), consistent with background cloud evident in Fig. 1. The Convective

Available Potential Energy (CAPE) as well as the maximum unstable CAPE (MUCAPE) during this period were zero, suggesting a lower potential for thunderstorm development. It is likely that much of the energy that lifted the air parcel and produced the pyroCb came from the fire itself rather than the unmodified atmosphere. By contrast, the sounding at a similar time (15:00 LT) on 9 January (when daily C-Haines was low during the fire period, Fig. S5) shows a temperature inversion occurring at around 850 hPa (1.3 km), with the air immediately above that level being moderately dry and stable (Fig. S2c).

[Figure]

**Fig. S2:** BARRA soundings for Hobart Airport on 4 and 9 January 2013 representing period of highest and lowest C-Haines respectively during the early days of the fire. P is atmospheric pressure in hPa. (a-b) Sounding at 15:00 and 16:00 LT on 4 January during rapid plume growth and pyroCb development, and (c) sounding at 15:00 on 9 January when smoke plume was not visible on the weather radar.

**S1.3 Temperature lapse rate at lower atmosphere**

Figure S2 shows a time series of air temperature lapse rate calculated from BARRA gridded air temperature at the 850 and
hPa pressure levels (or 1.3-5.5 km height) for January 2013 at Hobart Airport. The 850-500 hPa lapse rate gives an
indication of the (in)stability of the lower half of the troposphere. In Fig. S3, LR was highest at the start of the fire, peaking at
8.6°C km⁻¹ at 15:00 LT on 4 January 2013. LR >7.5°C km⁻¹ is considered very unstable (Peterson *et al.*, 2014) while LR of 6-
7.5 °C km⁻¹ is conditionally unstable, depending on the saturation level of the air. For saturated air parcels, >6 °C km⁻¹ is
unstable while for dry parcels, 6-7.5 °C km⁻¹ is stable. Convection is likely to be severe when the LR is above 7.5 °C km⁻¹.
Under such conditions, there is a likelihood of strong updrafts, thunderstorms and convective downdrafts, subsequently
increasing the severity of fire weather, and ultimately fire behaviour when surface conditions are elevated. Values of LR <6
are generally stable. In Fig. S3, beyond 4 January, the lower half of the troposphere fluctuated between stable and conditionally
unstable (from 5-31 January), with the afternoon (15:00) and night-time (21:00) having higher lapse rates than in the morning
(09:00) for most days, as is typically the case.

[Figure]

**Fig. S3:** Time series of temperature lapse rate between the 850-500 hPa levels for 09:00, 15:00 and 21:00 LT in January 2013
at the Hobart Airport.

**S2 Temporal smoke/pyroCb dynamics in Vorticity-driven Lateral Spread (VLS) prone areas**

[Figure]

**Fig. S4:** A map of VLS-prone areas within the Forcett-Dunalley fireground overlaid with smoke plume progression and DEM/contours during the period (15:00-17:30 LT on 4 January) that includes peak fire behaviour. The dashed lines in panel (a) represent a southwest (2 km) expansion of the upwind edge of the plume perpendicular to the prevailing northwest winds.

**S3 Time series fire weather (C-Haines & FFDI) during January 2013, including the Forcett-Dunalley fire period**

[Figure]

**Fig. S5:** Time series of daily maximum FFDI (calculated from 30-min weather data: air temperature, relative humidity, wind speed and soil moisture at Hobart Airport AWS) and daily maximum C-Haines (calculated from gridded BARRA model for Hobart Airport) in January 2013. Black vertical lines represent the start and end dates of the Forcett-Dunalley fire.

**S4 Time series of elevated fire weather and area burnt per fire management area (FMA) in Tasmania**

[Figure]

**Fig. S6:** Interannual variation of elevated fire weather days and area burnt in each fire management region in Tasmania. The bar graph represents number of days annually with elevated regional means of daily FFDI and C-Haines exceeding the following thresholds (FFDI>15 & C-Haines >7; and FFDI>25 & C-Haines >9). The line graph represents annual area burnt by wildfires irrespective of the ignition source.

**S5 Likely pyroCb pathway during the period of violent pyroconvection in Forcett-Dunalley fire**

[Figure]

**Fig. S7**: Blow-Up Fire Outlook (BUFO) model pathway for parts of the Forcett-Dunalley fire during the period of violent pyroconvection on 4 January 2013, adapted from McRae *et al.* (2018).